# DeNOTS: Stable Deep Neural ODEs for Time Series

**Ilya Kuleshov**[*]
Applied AI Institute
Moscow, Russia

**Evgenia Romanenkova**
Applied AI Institute
Moscow, Russia

**Vladislav Zhuzhel**
Applied AI Institute
Moscow, Russia

**Galina Boeva**
Applied AI Institute
Moscow, Russia

**Evgeni Vorsin**
Innotech
Moscow, Russia

**Alexey Zaytsev**
Applied AI Institute
Moscow, Russia

## Abstract

Neural Controlled Differential Equations (Neural CDEs) provide a principled framework for modelling irregular time series in continuous time. Their number of function evaluations (NFEs) acts as a natural analogue of depth in discrete neural networks and is typically controlled indirectly via solver tolerances. However, tightening tolerances increases numerical precision without necessarily improving expressiveness. We propose a simple alternative: scaling the integration time horizon to increase NFEs and thereby "deepen" the model. Since enlarging the interval can cause uncontrolled growth in standard vector fields, we introduce a Negative Feedback (NF) mechanism that ensures provable stability without limiting flexibility. We further establish general risk bounds for Neural CDEs and quantify discretization error using Gaussian process theory, improving robustness to integration and interpolation error. On four public benchmarks, our method, **DeNOTS**, outperforms existing approaches—including Neural RDEs and state space models—by up to $20\%$. DeNOTS combines expressiveness, stability, and robustness for reliable continuous-time modelling.

## 1 Introduction

Neural Controlled Differential Equations (CDEs) Kidger et al. (2020) provide a natural way to process irregular time series. CDEs are Ordinary Differential Equations (ODEs), where the derivative depends on an external input signal. Neural CDEs utilize a Neural Network (NN) as the ODE's dynamics function. It is well-known that increasing NN depth (number of layers) leads to higher expressiveness Gripenberg (2003); Lu et al. (2017); Yarotsky (2017), i.e., widens the class of functions the NN may represent Gühring et al. (2020). According to the original Neural ODE paper Chen et al. (2018), the natural analogue of NN depth is the number of function evaluations (NFE). Naturally, we hypothesise that a larger NFE results in a better Neural CDE model.

NFE is primarily controlled by the solver tolerance, which defines the acceptable error level during numerical integration Dormand & Prince (1980). Lowering this tolerance increases the required number of integration steps and NFE. However, prior work mostly sidesteps this topic: the expressiveness gains from higher precision are often minimal in practice. The problem here is that, as we will show, boosting expressiveness on a fixed integration interval necessitates larger $l_2$ weight norms, which harms training stability. Instead, we propose scaling the integration time. This method improves expressiveness while reducing the required weight norms. We refer to it as Scaled Neural CDE (SNCDE).

Upon investigating the proposed time-scaling procedure, we found that longer integration intervals can introduce uncontrollable trajectory growth, which must also be addressed. A very intuitive approach to constraining the trajectory is adding Negative Feedback (NF). Prior work implemented it by subtracting the current hidden state from the dynamics function De Brouwer et al. (2019).

---

[*]Corresponding author: i.kuleshov@applied-ai.ru.

Table 1: Properties comparison for SNCDEs with different vector fields.

| SNCDE Vector Field | Stability (Th. 4.4/Fig. 4) | Error bounds (Th. 4.5/Tab. 5) | Long-term memory (Th. 4.8/Tab. 4) |
|---|---|---|---|
| Unstable (No NF/Tanh/ReLU) | No | No | Yes |
| Sync-NF (orig. GRU-ODE) | Yes | Yes | No |
| DeNOTS (Anti-NF, ours) | Yes | Yes | Yes |

However, as we demonstrate, such a technique causes "forgetfulness": the influence of older states constantly decays, and the model cannot retain important knowledge throughout the sequence. This effect is akin to the one experienced by classic Recurrent Neural Networks (RNN), remedied by Long Short-Term Memory (LSTM) Hochreiter & Schmidhuber (1997) and Gated Recurrent Units (GRU) Chung et al. (2014). We demonstrate that our novel NF does not suffer from "forgetfulness".

Neural differential equation-based models possess another crucial property: we cannot calculate the solution perfectly, forcing us to approximate the true continuous process via discrete steps. We have to discretise the time series, so the models cannot analyse the system during the resulting inter-observation gaps, accumulating epistemic uncertainty in the final representation Blasco et al. (2024). Estimating such uncertainty has been a topic of interest for decades Golubev & Krymova (2013). Differential equation solvers also introduce discretisation errors, since they simulate a difference equation instead of the continuous differential one Hairer et al. (1993). Without additional modifications, the error accumulates with time, and the theoretical variance of the final prediction becomes proportional to sequence length.

To overcome the aforementioned challenges, this work introduces DeNOTS — Stable Deep Neural ODEs for Time Series. Instead of lowering tolerances, we scale the time interval. Larger time frames destabilise integration, so we include a novel Negative Feedback (NF) mechanism. Our NF enables stability and better expressiveness while maintaining long-term memory. Moreover, we prove that the discretisation uncertainty does not accumulate in DeNOTS's final hidden state: it is $\sim \mathcal{O}(1)$ w.r.t. the number of observations. Figure 1 compares various approaches to increasing NFEs for expressivity on a synthetic dataset.

Overall, our main contributions are as follows:

- The idea of deliberately scaling integration time to improve the model's representation power. We show that lowering tolerances for expressiveness does not adequately address theoretical limitations, providing minimal gains. SNCDE naturally circumvents these limitations, thus enhancing the representation power of the model and significantly boosting metrics.

- The antisynchronous NF mechanism (updates and NF are activated in anti-phase), which stabilises hidden trajectories on larger time scales but keeps the model flexible in practice. All alternatives have crucial flaws, as summarized in Table 1: non-stable dynamics are challenging to train, and synchronous NF is too restricitve, and tends to "forget" important information.

- Theoretical analysis of the effect of discretisation on our model, certifying that the numerical error does not accumulate in our prediction. We also provide a tight analytical bound for the interpolation risk of natural cubic splines.

- A quantitative experimental study, comparing DeNOTS to modern baselines on four open datasets. We demonstrate that our method excels in all settings.

In Section 2 we introduce the general framework of our method. Sections 3, 4 discuss the main contributions of our approach in detail. Section 5 provides our experimental results. Section 6 sums up our paper. We moved the Related Works section to Appendix A to save space.

## 2 METHOD

**Representation learning for non-uniform time series.** We focus on solving downstream tasks for time series with global, sequence-wise targets (binary/multiclass/regression). Let $S = \{(t_k, \mathbf{x}_k)\}_{k=1}^n$

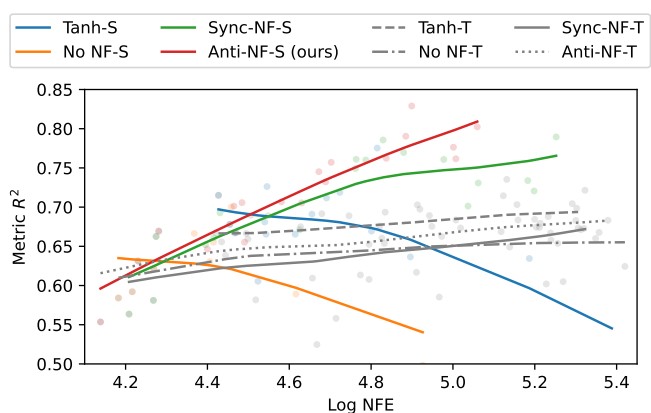

Figure 1: $R^2$ vs Log-NFEs on the Pendulum dataset for various methods of increasing NFEs: -T for lowering tolerance, -S for increasing time scale; for various vector fields (VF): Tanh — MLP with $\tanh$ activation; No NF — vanilla GRU VF, Sync NF — GRU-ODE VF, Anti NF — our version. The curves were drawn via Radial Basis Function interpolation.

be the analysed sequence, where $t_1 < t_2 < \ldots < t_n \in \mathbb{R}$ are the time stamps and $\mathbf{x}_k \in \mathbb{R}^u$ are the feature-vectors of the corresponding observations. The task is to predict the correct target from $S$.

For convenience, we assume $t_1 = 0$, and denote $t_n \triangleq T$. The sequence $S$ is passed through the backbone to obtain an embedding $\mathbf{h} \in \mathbb{R}^v$, which, supposedly, characterises $S$ as a whole. Finally, a linear head is used to transform $\mathbf{h}$ into the prediction $\hat{y}$.

**Neural CDEs.**  Neural CDEs provide an elegant way to deal with non-uniform time series. Formally, these methods integrate the following Cauchy problem for specific initial conditions $\mathbf{h}_0$, vector field (VF) $\mathbf{g}_\theta$, and an interpolation of the input sequence $\hat{\mathbf{x}}(t)$:

$$\text{Dynamics:} \begin{cases} \mathbf{h}(0) = \mathbf{h}_0; \\ \frac{d\mathbf{h}(t)}{dt} = \mathbf{g}_\theta(\hat{\mathbf{x}}(t), \mathbf{h}(t)). \end{cases} \qquad \text{Solution: } \mathbf{h}(t) : [0, T] \to \mathbb{R}^v. \qquad (1)$$

The output of our backbone is $\mathbf{h} \triangleq \mathbf{h}(T)$. We use a modified GRU cell Chung et al. (2014) for $\mathbf{g}_\theta$, and, following Kidger et al. (2020), interpolate the input data using natural cubic splines.

**The DeNOTS approach:**  schematically presented in Figure 2, it consists of the following steps.

1. Preprocess the input sequence with **time scaling**: $t_k \leftarrow \frac{D}{M} t_k$, where $D, M$ are hyperparameters.

2. Interpolate the multivariate features $\{\mathbf{x}_k\}_k$ to get $\hat{\mathbf{x}}(t), t \in [0, T]$ using cubic splines Kidger et al. (2020).

3. Integrate the Neural ODE with **Negative Feedback** (Anti-NF) starting from $\mathbf{h}_0$ over $\hat{\mathbf{x}}(t)$ to get $\mathbf{h} \triangleq \mathbf{h}(T)$.

4. Pass the final hidden state $\mathbf{h}$ through the linear layer to get the prediction $\hat{y}$.

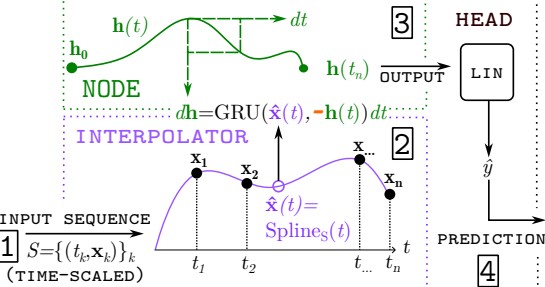

Figure 2: Scheme of the proposed DeNOTS method. The large red minus sign ($-$) represents our NF.

Our contributions revolve around two key ideas: time-scaling and a novel anti-phase negative feedback (Anti-NF) mechanism, which jointly address the limitations of existing Neural CDE approaches. In the following sections, we present a detailed description of the DeNOTS framework and the corresponding theoretical analysis supplemented with proofs in Appendix B. To aid clarity, we illustrate each theoretical result with numerical experiments.

## 3   TIME SCALING

As mentioned in the introduction, we argue that scaling time benefits expressiveness. By expressiveness, we mean the broadness of the class of functions that our network can represent. Now, we introduce several definitions.

**Differentially definable mappings from trajectories $\mathcal{F}$.** We say that $F_{\mathbf{g}} \in \mathcal{F}$, if there exists an ODE with the vector field $\mathbf{g}(\mathbf{x}(t), \mathbf{h}(t)) : \mathbb{R}^u \times \mathbb{R}^v \to \mathbb{R}^v$, with the starting condition $\mathbf{h}(0) = \mathbf{h}_0$ and the solution $\mathbf{h}(t) : \mathbb{R} \to \mathbb{R}^v$: $\frac{d}{dt}\mathbf{h}(t) = \mathbf{g}(\mathbf{h}(t), \mathbf{x}(t))$, such that $F_{\mathbf{g}}(\mathbf{x}(\cdot), \mathbf{h}_0; t) = \mathbf{h}(t)$. Not all functions on trajectories can be represented this way.

**Start-corrected differentially-definable mappings,** i.e. the elements of $\mathcal{F}$, minus their initial state $\mathbf{h}_0$: $\mathring{\mathcal{F}} = \{\mathring{F}_{\mathbf{g}}(\mathbf{x}(\cdot), \mathbf{h}_0; t) = F_{\mathbf{g}}(\mathbf{x}(\cdot), \mathbf{h}_0; t) - \mathbf{h}_0 | F_{\mathbf{g}} \in \mathcal{F}\}$. Since we set $\mathbf{h}_0 = 0$ in most practical cases, the distinction is rather technical; it is here mostly to account for Lipschitzness w.r.t. $\mathbf{h}$, as we will see below.

**Lipschitz constraints.** We introduce two ways to constrain this class with Lipschitz constants: with $\mathbf{g}$ being $M_x, M_h$-Lipschitz

$$\mathring{\mathcal{F}}_g(M_x, M_h) = \{\mathring{F}_{\mathbf{g}} \in \mathring{\mathcal{F}} | \mathbf{g} - M_x, M_h\text{-lipschitz w.r.t. } \mathbf{x}, \mathbf{h}\},$$

and with $\mathring{F}_{\mathbf{g}}(\dots; t)$ being $L_x(t), L_h(t)$-lipschitz, where $L_x(t), L_h(t) : \mathbb{R} \to \mathbb{R}_+$:

$$\mathring{\mathcal{F}}_F(L_x(\cdot), L_h(\cdot)) = \{\mathring{F}_{\mathbf{g}} \in \mathring{\mathcal{F}} | \mathring{F}_{\mathbf{g}} - L_x(t), L_h(t)\text{-lipschitz w.r.t. } \mathbf{x}(\cdot), \mathbf{h}_0\},$$

where Lipschitzness w.r.t. $\mathbf{x}(\cdot)$ is meant in terms of $L_2$ function norms: $\|\mathbf{x}(\cdot)\|_{L_2} = \sqrt{\int \|\mathbf{x}(t)\|_2^2 dt}$.

Intuitively, $\mathring{\mathcal{F}}_g(M_x, M_h)$ represents all mappings that our NCDE can learn, given a Lipschitz-constrained Neural Network $\mathbf{g}$, while $\mathring{\mathcal{F}}_F(L_x(\cdot), L_h(\cdot))$ represents all the possible Lipschitz-mappings that we may want to learn.

Equipped with the above definitions, we can formulate the following:

**Theorem 3.1** (Expressiveness). *The classes $\mathring{\mathcal{F}}_g(M_x, M_h)$ and $\mathring{\mathcal{F}}_F(L_x(\cdot), L_h(\cdot))$ are equal, given the following relations between their arguments:*

$$L_x(t) = M_x \sqrt{\frac{1}{2M_h}\left(e^{2M_h t} - 1\right)}; L_h(t) = e^{M_h t} - 1.$$

As a corollary, given an $M_x, M_h$-Lipschitz NN $\mathbf{g}$, we can represent all the $L_x, L_h$-Lipschitz (differentially-definable, start-corrected) $F$, and only them. Consequently, one way to boost expressivity is to increase $M_h$, corresponding to the weights' norms. Larger weight norms induce saturation for bounded activations such as tanh or sigmoid Ven & Lederer (2021) or "dying" for unbounded activations such as ReLU Lu et al. (2019), destabilising training in both cases. Alternatively, scaling $T$ makes the model exponentially more expressive without increasing the magnitude of weights. We refer to Neural CDEs with time scaling as *"Scaled Neural CDEs"* (**SNCDEs**). Formally, time scaling means we set $t_k \leftarrow \frac{D}{M} t_k$, where $D > 0$ is a hyperparameter and $M > 0$ is a dataset-specific normalising constant, introduced to make values of $D$ comparable across benchmarks.

According to Theorem 3.1, if we fix $L_F$ to the desired value, a larger $D$ allows for a lower $M_h$. Figure 3 illustrates that increasing $D$ indeed lowers the $l_2$ norms of weights. Furthermore, we empirically validate the stabilising benefits of time scaling for a synthetic task of classifying 1D functions into bumps or constant-zero ones. As demonstrated by Table 2, models with larger time scales perform better than ones with lower tolerances[1]. Further experiments on this topic are in Section 5 and Appendix C.6.

## 4 NEGATIVE FEEDBACK

DeNOTS also modifies the form of the dynamics function: it carefully pushes the hidden trajectory towards zero by including *negative feedback* in the derivative. NF provides numerous benefits, which we motivate in theory and with practical experiments on synthetic data, as summarized in Table 1.

---

[1]This is clearly observed for models with Tanh activations, but less so for ReLU-activated ones, which fail to reach perfect quality: likely due to decreased training stability of unlimited vector fields, amplified by time scaling.

Table 2: AUROC for Bump synthetic dataset results for two vector fields: Tanh-activated and ReLU-activated MLPs. The columns represent different ways to increase NFE: "Default" – with no modifications, "Tolerance" – for lowered tolerances, and "Scale" – for increased time scale.

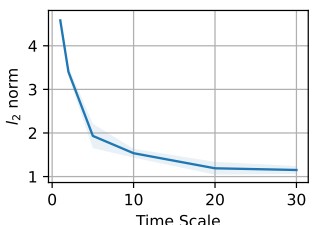

| VF | Default | Tolerance | Scale |
|------|-----------------|-----------------|-----------------|
| Tanh | $0.77 \pm 0.02$ | $0.90 \pm 0.01$ | $\mathbf{0.99 \pm 0.00}$ |
| ReLU | $\mathbf{0.89 \pm 0.01}$ | $0.85 \pm 0.01$ | $\mathbf{0.91 \pm 0.05}$ |

Figure 3: $l_2$ norms of weights vs time scale $D$ for the $\texttt{tanh}$ vector field on the Pendulum dataset.

We model NF by the following differential equation, a specific case within the dynamics $(1)^2$:

$$\frac{d\mathbf{h}(t)}{dt} = \mathbf{g}_\theta(\hat{\mathbf{x}}(t), \mathbf{h}(t)) = a\mathbf{f}_\theta(\hat{\mathbf{x}}(t), \mathbf{h}(t)) - b\mathbf{h}(t), a, b \in \mathbb{R}. \qquad (2)$$

The values of $a$ and $b$ determine the relative magnitude of the NF term and the overall scale of the derivative. For clarity, our theoretical analysis focuses on scalar-valued $a$ and $b$, which suffice to demonstrate the key properties. Extending the analysis to vector-valued parameters introduces substantial technical complexity without offering additional insight.

To conduct our analysis, we need to constrain the function $\mathbf{f}_\theta$ and the values $a, b, L_h$:

**Assumption 4.1.** (1) The function $\mathbf{f}_\theta$ from (2) is Lipschitz w.r.t. both $\hat{\mathbf{x}}$ and $\mathbf{h}$ with constants $L_x, L_h$ correspondingly, and (2) it is equal to zero for zero vectors $\mathbf{f}_\theta(\mathbf{0}_u, \mathbf{0}_v) = \mathbf{0}_v$.

**Assumption 4.2.** The values of $a, b, L_h$ are constrained by the following:

$$a \in (0, 1), b \in (0, 1), L_h \in (0, 1), aL_h < b. \qquad (3)$$

In practice, the right-hand side of 2 is a slightly modified version of the classic GRU architecture Chung et al. (2014). Specifically, we work with the last step in the GRU block, which is:

$$\tilde{\mathbf{h}} = \mathrm{GRU}(\hat{\mathbf{x}}, \mathbf{h}(t)) = (1 - \mathbf{z}) \odot \mathbf{n} + \mathbf{z} \odot \mathbf{h}, \qquad (4)$$

where $\mathbf{z}$ is the output of a sigmoid-RNN layer, $\mathbf{n}$ is the output of a tanh-RNN layer from previous steps of the GRU, and $\tilde{\mathbf{h}}$ is the final output of the GRU layer. The full specification of the GRU architecture is located in Appendix C.2.

We consider three NF options, with their names, derivation from the GRU (4), and the corresponding vector fields provided in columns 1-3 of Table 3. Rewriting each of these variants in terms of (2), we get three individual sets of constraints for $a, b$, presented in the fourth column of Table 3. Next, we substitute these constraints into (3) in the last column of Table 3. We analyse whether Assumption 4.2 stands in practice for Sync-NF and Anti-NF in Appendix C.7. Each considered variant of SNCDE with NF is described in detail below.

**No NF.** The GRU model is used as-is for the vector field of Neural CDE[3]. The combination of (4) with (2) implies $b < 0$, i.e. *positive feedback*. Assumption 4.2 never stands for No-NF, since $b < 0$. We later show that such a vector field severely destabilises the trajectory on larger values of $D$ (see Section 4.1).

**Sync-NF.** The GRU-ODE version, proposed in De Brouwer et al. (2019), which subtracts $\mathbf{h}$ from (4). Here, both the update term $\mathbf{n}$ and the NF term $-\mathbf{h}$ are activated simultaneously, with $\mathbf{z}$ affecting only the overall scale of the derivative, so we refer to this version as *Sync-NF*. The $l_2$ norms of the weights are smaller on longer integration intervals (as we concluded in the previous section), so

---

[2]Note, that setting $a = 1; b = 0$ brings us back to the general NN-defined vector field (1), so this constraint does not limit applicability. The real constraint is introduced by Assumption 4.2, but our Anti-NF does not necessarily adhere to it, as analysed in Section 4.3 and Appendix C.7

[3]From here on: GRU refers to the recurrent network; the SNCDE with GRU as the dynamics is "No NF".

Table 3: Various options of GRU-based NF, with their derivations from (4) (col. 2), the corresponding vector fields (col.3), $a, b$ constraints in terms of (2) (col. 4), and the resulting form of Assumption 4.2.

| Name | Derivation from (4) | Vector Field, $\mathbf{g}_\theta$ | $a, b$ constr. | Assumption 4.2 |
|---|---|---|---|---|
| No NF | $\text{GRU}(\hat{\mathbf{x}}(t), \mathbf{h}(t))$ | $(1-\mathbf{z}) \odot \mathbf{n} + \mathbf{z} \odot \mathbf{h}$ | $a - b = 1$ | (✗) |
| Sync-NF | $\text{GRU}(\hat{\mathbf{x}}(t), \mathbf{h}(t)) - \mathbf{h}(t)$ | $(1-\mathbf{z}) \odot (\mathbf{n} - \mathbf{h})$ | $a - b = 0$ | (✓), $L_h < 1$ |
| Anti-NF | $\text{GRU}(\hat{\mathbf{x}}(t), -\mathbf{h}(t))$ | $(1-\mathbf{z}) \odot \mathbf{n} - \mathbf{z} \odot \mathbf{h}$ | $a + b = 1$ | ($\sim$), $aL_h < 1 - a$ |

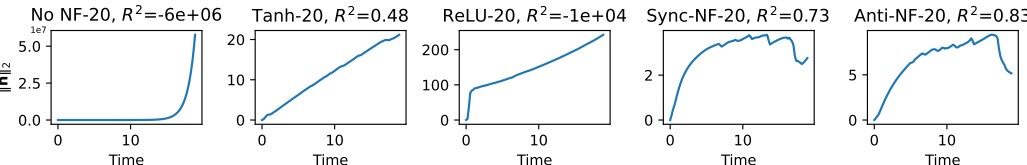

Figure 4: Trajectories for various vector fields on the Pendulum dataset, with $D = 20$. The corresponding $R^2$ values on test are provided in the title above each graph.

Assumption 4.2 always stands for Sync-NF. The Sync-NF version without time scaling ($D = 1$) is equivalent to GRU-ODE[4] De Brouwer et al. (2019). As its authors prove, the corresponding trajectory is constrained to $[-1, +1]$. Moreover, it does not regulate the scale of NF relative to the update, implying that, intuitively, the model may "forget" prior knowledge (see Section 4.3).

**Anti-NF (DeNOTS, ours).** We pass $-\mathbf{h}$ instead of $\mathbf{h}$ to the GRU layer. In our version, the NF and update terms are activated *separately*, with $\mathbf{z}$ regulating the scale of NF relative to the update, so we refer to this version as *Anti-NF*. The validity of (3) for Anti-NF depends on the value of $a$. Consequently, our model can disable the "forgetting" effect by increasing $a$, which is learnable and even adaptive, since it corresponds to $\mathbf{z}$ from the GRU (4). Setting $a = 1, b = 0$ lifts the NF constraint altogether, turning (2) into a generic Neural CDE: $\frac{d\mathbf{h}}{dt} = \mathbf{f}_\theta$.

## 4.1 STABILITY

A crucial issue to consider when working with ODEs is stability analysis Boyce et al. (1969); Oh et al. (2024). On longer time frames, unstable ODEs may have unbounded trajectories even with reasonable weight norms, which is detrimental to training due to saturation Ioffe & Szegedy (2015) or "dying" Lu et al. (2019) of activations. The equation (1) depends on an input signal $\hat{\mathbf{x}}(t)$, so we approach stability analysis in terms of control systems theory Sontag & Wang (1995):

**Definition 4.3.** We call the ODE (1) *input-to-state stable*, if there exist two continuous functions: an increasing $\gamma(\cdot)$ with $\gamma(0) = 0$, and $\beta(\cdot, \cdot)$, increasing in the first argument, $\beta(0, \cdot) = 0$ and strictly decreasing in the second argument, $\beta(\cdot, t) \underset{t \to \infty}{\to} 0$, such that:

$$\|\mathbf{h}(t)\|_2 \leq \beta(\|\mathbf{h}_0\|_2, t) + \gamma(\|\hat{\mathbf{x}}\|_\infty). \tag{5}$$

Intuitively, equation (5) means that for any bounded $\hat{\mathbf{x}}$ the trajectory remains bounded and stable.

**Theorem 4.4.** *The ODE* (2) *under assumptions 4.1, 4.2 is input-to-state stable.*

Thus, the Sync-NF and Anti-NF models are stable. We test the benefits of Theorem 4.4 in practice, presenting the trajectory norms $\|\mathbf{h}(t)\|_2$ for $D = 20$ along with the metrics in Figure 4. Non-stable vector fields, such as No-NF and MLP-based ones with ReLU and Tanh activations, induce uncontrolled growth and achieve poor $R^2$ due to unstable training. The tanh-activated vector field is less prone to instability, since the corresponding derivative is bounded. The SNCDE remains stable for both Sync-NF and Anti-NF, and the corresponding metrics are significantly better. Notably, Anti-NF slightly outperforms Sync-NF: we argue this is due to a more flexible trajectory.

---

[4]From here on: GRU-ODE refers to the corresponding NCDE without time scaling ($D = 1$).

## 4.2 ERROR BOUNDS

Until now, we have assumed that the Neural Differential Equation (1) which our method models can be solved perfectly. Unfortunately, due to physical restrictions of our computational devices, obtaining the perfect solution is impossible. We are forced to discretise the continuous processes that we work with, which introduces certain errors into our calculations:

- Discretisation of the input process $\hat{\mathbf{x}}(t)$ introduces the *interpolation error*. The interpolation error is a property of the interpolation routine.
- Discretisation of the numerical integration process introduces the *integration error*. The integration error is a property of the differential equation solver.

It is a well-known result Hairer et al. (1993), that the global numerical integration error is exponential w.r.t. time. In a sense, we observe precisely that in Figure 3, where the hidden norms explode for non-stable vector fields. The exponential accumulation of error significantly hampers the time-scaling routine that we propose, so training such models becomes close to impossible. Fortunately, our stable Anti-NF and Sync-NF models, adhering to Assumption 4.2, do not accumulate the numerical error in their final hidden state.

To devise the theoretical results of this section, we need to introduce the "perfect" Neural CDE, calculated without any errors:

$$\mathbf{h}^*(0) = \mathbf{h}_0; \quad \frac{d\mathbf{h}^*(t)}{dt} = a\mathbf{f}_\theta(\mathbf{x}(t), \mathbf{h}^*(t)) - b\mathbf{h}^*(t). \tag{6}$$

Then, both the integration and the interpolation errors may be modelled via the following "imperfect" ODE, often referred to as the "Modified Vector Field" Reich (1999):

$$\mathbf{h}(0) = \mathbf{h}_0; \quad \frac{d\mathbf{h}(t)}{dt} = a\mathbf{f}_\theta(\mathbf{x}(t), \mathbf{h}(t)) - b\mathbf{h}(t) + \xi(t), \tag{7}$$

where $\xi(t)$ is a random vector, which represents the momentary deviation of our numerical ODE from the true one. For further theoretical considerations, we introduce two types of bounds on $\xi(t)$:

$$\text{Pointwise: } \mathbb{E}\|\xi(t)\|_2^2 \leq \xi_{\text{PW}}^2, \forall 0 \leq t \leq T; \quad \text{Interval: } \frac{1}{T}\int_0^T \mathbb{E}\|\xi(t)\|_2^2 \, dt \leq \xi_{\text{INT}}^2. \tag{8}$$

For each of the introduced bounds, we can devise an estimate for the error in the final hidden state:

**Theorem 4.5.** *Error Bounds (Robustness). Consider the differential equations* (6),(7)*, under Assumptions 4.1, 4.2 Then, depending on whether we are working with the pointwise or the interval errors* (8)*, we can write different bounds on the variance, accumulating in* $\mathbf{h}$.

*Pointwise bound:* 
$$\mathbb{E}\|\mathbf{h}(T) - \mathbf{h}^*(T)\|_2^2 \leq \left(\frac{a}{b - aL_h}\right)^2 \xi_{\text{PW}}^2. \tag{9}$$

*Interval bound:* 
$$\frac{1}{T}\int_0^T \mathbb{E}\|\mathbf{h}(t) - \mathbf{h}^*(t)\|_2^2 \, dt \leq \left(\frac{a}{b - aL_h}\right)^2 \xi_{\text{INT}}^2. \tag{10}$$

The errors (9) and (10) are independent of the total integration time $T$. Consequently, the error *does not accumulate in our final hidden state*. Additional analysis of tightness of (9), (10) is in Appendix B.4.1.

Numerical integration error is directly connected to our momentary deviation $\xi(t)$, and, as we mentioned above, the benefits of NF for solution stability are already validated by Figure 3. However, the relation of interpolation error to $\xi$ is less straightforward, we discuss it in the following section.

### 4.2.1 INTERPOLATION ERROR

Say, $\hat{\mathbf{x}}(t)$ is the cubic spline interpolation of the input time series. For analytic purposes, we also introduce the true (unobserved) value of the input process, $\mathbf{x}(t)$. Similar to (8), we consider two types of interpolation errors:

$$\|\hat{\mathbf{x}}(t) - \mathbf{x}(t)\|_2^2 \leq \sigma_{\text{PW}}^2, \quad \int_{t_k}^{t_{k+1}} \|\hat{\mathbf{x}}(t) - \mathbf{x}(t)\|_2^2 \, dt \leq \sigma_{\text{INT}}^2. \tag{11}$$

Given the absence of other errors, the momentary deviation $\xi(t)$ is connected to $\|\mathbf{x} - \hat{\mathbf{x}}\|$ via the $L_x$ Lipschitz coefficient, with $\xi_{\text{PW/INT}} = L_x \sigma_{\text{PW/INT}}$, so we may focus on estimating the $\sigma$-errors. The bound for the pointwise error of cubic spline interpolation is a well-known result De Boor & De Boor (1978):

$$\|\mathbf{x}(t) - \hat{\mathbf{x}}(t)\| \leq \frac{5}{384} \delta_{\max}^4 \|\mathbf{x}^{(4)}(t)\|, \tag{12}$$

where $\mathbf{x}^{(4)}$ is the fourth derivative of the input trajectory, and $\delta_{\max}$ is the maximum step between observations. Assuming that $\|\mathbf{x}^{(4)}\|$ is bounded, we can plug (12) directly into (8) (remebering to include $L_x$).

However, the bound (12) is not tight due to the nature of cubic splines. The bound may correctly approximate the error in the middle of inter-observation gaps, but the error decreases to zero as we move towards each observation. For a more precise point of view, we can estimate the interval error from (11). We base our interval error estimation on Gaussian Process (GP) theory: the authors of Golubev & Krymova (2013) proved that, under certain conditions, cubic splines can be seen as a limit case of Gaussian Process regression. Using an approach, similar to that of Zaytsev & Burnaev (2017), we were able to devise a tight bound for such a Gaussian Process limit, which we provide in the following lemma:

**Lemma 4.6.** *For some $\xi > 0, Q > 0$, say that $\hat{\mathbf{x}}(t)$ is a Gaussian Process with the spectral density* [5]: $F_i(\omega) = \frac{Q}{\omega^4 + \xi^4}$. *Also, assume that the considered sequence $S$ is infinite, and the GP covariance function is such that increasing interval sizes increases the expectation of the interval error from (11). Then, the interval error $\mathbb{E}\sigma_{INT}^2(\xi)$ has the following asymptotic:*

$$\frac{4\pi^4 \sqrt{3}}{63} u Q \delta_{\min}^4 \leq \lim_{\xi \to 0} \mathbb{E}\sigma_{INT}^2(\xi) \leq \frac{4\pi^4 \sqrt{3}}{63} u Q \delta_{\max}^4.$$

A full version of the reasoning behind the above Lemma is provided in the Appendix.

**Corollary 4.7.** *Plugging the above into (10), we get the tight upper bound for cubic splines $\hat{\mathbf{x}}(t)$:*

$$\frac{1}{n} \int_0^T \mathbb{E}\|\mathbf{h}(t) - \mathbf{h}^*(t)\|_2^2 \, dt \leq \left(\frac{a L_x}{b - a L_h}\right)^2 \frac{4\pi^4 \sqrt{3}}{63} u Q \delta_{\max}^4.$$

To test the effect of interpolation error on our models, we perform drop attacks (dropping a fraction of input tokens and replacing them with NaNs) on the Pendulum dataset. According to Table 5 (first row), all SNCDEs are robust to drop attacks, with NF versions outperforming No NF. The Sync-NF model is slightly better than Anti-NF, likely due to a larger $b - a L_h$, but that difference is insignificant compared to other baselines.

## 4.3 LONG-TERM MEMORY

Despite the implications of Theorems 4.4,4.5, strong negative feedback might not always be beneficial. We found that its usage may also induce "*forgetting*" – the model may lose important prior knowledge.

Unfortunately, the concept of "importance" may vary dataset-to-dataset, or even sample-to-sample, a good model should adaptively pick out such knowledge on its own, so defining forgetfulness is a difficult task. That said, we opt for a simpler definition, which generally aligns with our understanding: a model is "forgetful", if the influence of older trajectory parts constantly decays. Such decay leads to limited long-term memory of previous states.

Equipped with the above definition, we can prove that a strict negative feedback induces such forgetfulness:

**Theorem 4.8.** *For the ODE (2) under Assumptions 4.1, 4.2 it holds that:* $\frac{d}{d\tau} \left\| \frac{d\mathbf{h}(t+\tau)}{dh_i(t)} \right\|_2 < 0$.

The above implies that a shift in the beginning of the trajectory will not significantly influence the end of the trajectory. Fortunately, our Anti-NF model can adapt the relative values of $a, b$ to reduce or disable forgetfulness, making it more flexible. On the other hand, the Sync-NF model does not have this liberty: for $L_h < 1$, the Assumption 4.2 always holds, so it will always forget prior values. We further analyse this in Appendix C.7.

---

[5] At the limit $\xi \to 0$, the mean of the resulting GP tends to a natural cubic spline Golubev & Krymova (2013). The parameter $Q$ here represents our a priori beliefs about the data variance.

Table 4: $R^2$ on the SineMix dataset, averaged over five runs.

| RNN | GRU | Tanh | ReLU | No-NF | Sync-NF | Anti-NF |
|-----|-----|------|------|-------|---------|---------|
| -0.3 | 0.8 | 1 | 1 | 1 | 0.3 | 1 |

Table 5: The models' $R^2$ for C (change) and D (drop) attacks on Pendulum dataset. The suffix indicates the fraction of altered tokens: $0.85$ for Drop and $0.01$ for change. The columns represent the models: TF, RF for Temp- and RoFormer, NCDE/NRDE for the original Neural CDE/RDE models, and the last three columns present SNCDEs (Scaled Neural CDEs, $D = 5$) with three different vector fields. The full picture, given by the set of fractions vs metric curves, is provided in Appendix C.9.

| Attack | TF | RF | Mamba | NCDE | NRDE | GRU | No NF | Sync-NF | Anti-NF |
|--------|-----|-----|-------|------|------|-----|-------|---------|---------|
| D-0.85 | 0.16 | 0.31 | 0.36 | 0.02 | 0.20 | 0.27 | 0.40 | 0.64 | 0.61 |
| C-0.01 | -0.29 | -0.29 | -0.46 | -0.97 | -0.11 | 0.37 | 0.51 | 0.61 | 0.64 |

To illustrate the forgetting effect, we conduct experiments on a synthetic Sine-Mix dataset to compare Anti-NF's forgetfulness to Sync-NF's and classic RNNs. Each sequence consists of two equal-length sine waves with different frequencies, joined continuously at the mid-point. The target is to predict the frequency of the first wave, which is trivial unless the model forgets the beginning of the sequence. The results are in Table 4. After 5 training epochs, the Sync-NF version achieves only $R^2 = 0.3$, while the versions with all other vector fields, including Anti-NF, completed the task perfectly each time. Simultaneously, the vanilla RNN cell failed the task completely, while the GRU architecture was able to solve it rather well, indicating that Sync-NF's observed forgetfulness is akin to that of classic recurrent networks.

On the other hand, forgetfulness could make the model more robust: time series often evolve quickly, so forgetting irrelevant observations may be beneficial Cao et al. (2019). To gauge this effect, we measure our method's performance on perturbed sequences, with a small fraction replaced by standard Gaussian noise. The results are provided in Table 5 (row 2). Indeed, all the possibly forgetful models, i.e., the SNCDE versions and the GRU, are in the lead. Adding NF boosts robustness: Sync-NF and Anti-NF significantly outperform No-NF. Sync-NF performs slightly worse than Anti-NF. We hypothesise this is because the attack is uniformly distributed across the sequence: the corrupted tokens may appear closer to the end, in which case the Sync-NF model performs worse due to forgetfulness.

## 5 MAIN EXPERIMENTS

We tested the individual properties of DeNOTS in Sections 3, 4. Here, we present a general picture, measuring the quality of our method on downstream tasks, and extensively testing our main expressiveness claim. Due to space limitations, we moved the datasets' and models' descriptions, further details, and some additional experiments to Appendix C.

**Classification/Regression.** To test our model's overall performance, we compare it to other baseline models (including the Sync-NF vector field) on various tasks. The results are provided in Table 7. Our model is ranked first across all the considered datasets.

Table 6: Pearson correlation between log-NFE and the corresponding metric ($R^2$ for Pendulum and Sine-2, and AUROC for Sepsis), for various vector fields. In a pair $X/Y$, $X$ indicates the statistic for reducing tolerance, $Y$ — the statistic for increasing time scale. Values $\geq 0.7$ are highlighted in bold.

|        | Pendulum | Sepsis | Sine-2 |
|--------|----------|--------|--------|
| Tanh | 0.3/-0.6 | 0.3/**0.8** | 0.4/**0.7** |
| ReLU | 0.1/-0.6 | 0.3/-0.09 | 0.07/0.6 |
| No NF | 0.4/-0.5 | 0.3/-0.1 | 0.09/-0.5 |
| Sync-NF | 0.4/**0.8** | 0.5/-0.4 | 0.5/**1.0** |
| Anti-NF | 0.5/**0.9** | 0.5/**0.8** | 0.4/**1.0** |

**Scale expressiveness.** Next, we extensively test our main claim: combining SNCDE with Anti-NF (DeNOTS) is the best way to increase the model's expressiveness. However, measuring expressiveness

Table 7: Table with performance ranks (lower is better), built on our results. The rank of each backbone is one plus the number of other backbones, the score of which is better according to a one-sided T-test on the corresponding mean and deviations, with the p-value cutoff set to 0.1. Therefore, lower is better, a rank of 1.0 means that there are no better baselines, and the one in question is the best.

| Backbone | UWGL | InsectSound | Pendulum | Sepsis | Mean |
|---|---|---|---|---|---|
| DeNOTS (ours) | 1 | 1 | 1 | 1 | 1.0 |
| Sync-NF SNCDE (ours) | 1 | 1 | 2 | 1 | 1.25 |
| TempFormer | 4 | 1 | 6 | 5 | 4.0 |
| Neural CDE | 1 | 8 | 2 | 5 | 4.0 |
| Neural RDE | 1 | 6 | 1 | 9 | 4.25 |
| RoFormer | 5 | 5 | 6 | 3 | 4.75 |
| GRU | 8 | 1 | 4 | 7 | 5.0 |
| Mamba | 6 | 1 | 6 | 8 | 5.25 |
| GRU-ODE | 8 | 7 | 6 | 3 | 6.0 |

seems non-trivial at first glance. Fortunately, in our setup, expressiveness is quantified by model performance, as we explain in Appendix C.5.

We measure the Pearson correlation coefficient between log-NFE and the metric value for increasing precision or scaling time with various vector fields — different approaches to increasing expressiveness. The results are presented in Table 6. Indeed, DeNOTS is the only model that reliably displays a strong correlation on all three datasets. Further demonstrations can be found in Appendix C.6.

## 6 CONCLUSIONS

We propose a novel approach to increasing Neural ODE expressiveness in the time series domain, backed by theoretical and empirical evidence. Instead of lowering tolerances, we scale the integration interval, boosting metrics while keeping the $l_2$ norms of weights low. However, time scaling destabilises conventional vector fields, so we modify the dynamics function to include Negative Feedback. NF brings provable stability and robustness benefits, which hold in practice. Although prior versions of NF also possess these qualities, they have issues with long-term memory due to less flexible dynamics and strict trajectory constraints. In contrast, our Anti-NF preserves expressiveness and avoids forgetting. Combined with time scaling, this results in superior performance across four open time series benchmarks, outperforming recent Neural RDEs and state space models.

**Limitations and Future Work.** Neural CDE methods are fundamentally sequential, as they represent continuous-time dynamics through stepwise evolution that resists full parallelization. This intrinsic characteristic results in longer runtimes compared to architectures built for parallel computation. In DeNOTS, time scaling amplifies this effect by increasing the number of required update steps, exchanging additional computation for greater expressive capacity. Crucially, DeNOTS delivers strong empirical results, partly justifying the added cost by converting deeper sequential processing into tangible performance gains. Nevertheless, future research on Neural CDEs should emphasize scalable designs that retain their expressive strengths while enabling more parallel execution—much like the field's transition from recurrent models to Transformers. Some inefficiency could also be attributed to a high-level implementation, which also needs work. Finally, it could be interesting to adapt the novel concept of scaling integration time to other domains, such as event sequences, text, images, or tabular data.

**Reproducibility statement.** Our code is available at `https://github.com/Ilykuleshov/denots_iclr2025`. It contains automated scripts to download all our datasets, and uses popular frameworks to make it recognisable to a wide audience of ML researchers, reproducing all our baselines in these frameworks. A detailed description of the datasets can be found in Appendix C.1. The baselines are described in Appendix C.2. More information about our pipeline can be found in Appendix C.3.

ACKNOWLEDGMENTS

The work was supported by the grant for research centers in the field of AI provided by the Ministry of Economic Development of the Russian Federation in accordance with the agreement 000000C313925P4F0002 and the agreement №139-10-2025-033.

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

APPENDICES

## A    RELATED WORKS

**Neural ODEs for time-series.**    Neural ODEs provide an elegant way to take the sequence irregularity into account Oh et al. (2025). The representation evolves simultaneously with the observed system Chen et al. (2018). For example, GRU-ODE-Bayes De Brouwer et al. (2019) used Bayesian methods to account for missing observations optimally. To constrain the hidden trajectory to $[-1, 1]$, its authors introduced a strict NF, subtracting the current hidden state from the derivative.

As an improvement, the authors of Kidger et al. (2020) proposed to combine Neural ODEs with the theory of CDEs Friz & Victoir (2010) for time series classification. Instead of embedding the input data into the starting point, Neural CDEs interpolate it, using the resulting function to guide the trajectory. On the other hand, the original Neural CDEs from Kidger et al. (2020) are not sensitive to the irregularity of time intervals. So, the authors append the time difference as a separate feature. Moreover, these models have high memory usage. The controlling NN has to generate a matrix, which implies a weight dimension of $\sim d^3$, where $d$ is the dimensionality of the input features, instead of $d^2$ for conventional models, such as RNNs. DeNOTS employs a GRU cell as its dynamics, making it more memory efficient than Neural CDE, and sensitive to input time intervals, which is crucial for scaling time.

**Stability.**    Neural ODE stability is a well-researched topic. Prior work has approached it through adding specialized loss terms Kang et al. (2021); Rodriguez et al. (2022), constraining the form of the vector field Massaroli et al. (2020a) or the eigenvalues of the NN weights Tuor et al. (2020). However, very little work has been done for Neural CDEs, since the problem is significantly complicated by the presence of an external "control signal". Most notably, Morrill et al. (2021b) processes the input sequence in a windowed fashion, aggregating each window into a certain set of features to reduce length, while Morrill et al. (2021a) highlights the importance of selecting a continuous interpolant for the well-posedness of the initial-value problem. We prove that our CDE-type model is input-to-state stable, providing a significantly more general and formal result.

**Time.**    To escape the instability, introduced by possibly large time frames, most prior works enforce time to be roughly [0, 1] by setting it directly Chen et al. (2018); Rubanova et al. (2019); Dupont et al. (2019), re-parametrizing it Chen et al. (2020) or predicting it within tight bounds Massaroli et al. (2020b). Others devise models invariant to time transforms Kidger et al. (2020). To the best of our knowledge, all existing works do not consider time to be an important hyperparameter.

**Gaussian process interpolation errors.**    Our theoretical analysis investigates discretisation uncertainty, which manifests as GP variance. This involves estimating the squared error of an integral over a linear function of a GP with multiple outputs. For a specific assumption, we have a reasonable estimate of the quadratic risk that follows from Golubev & Krymova (2013). While alternatives exist Stein (2012); van der Vaart & van Zanten (2008); Zaytsev et al. (2018), they are unsuitable for our setting, with an additional ODE layer on top.

## B    THEORY

To enhance the reader experience, we duplicate all relevant statements and discussions from the main body in the Appendix, preserving reference numbering.

### B.1    TIME SCALING

As mentioned in the introduction, we argue that scaling time benefits expressiveness. By expressiveness, we mean the broadness of the class of functions that our network can represent. Now, we introduce several definitions.

**Differentially definable mappings from trajectories $\mathcal{F}$.**    We say that $F_{\mathbf{g}} \in \mathcal{F}$, if there exists an ODE with the vector field $\mathbf{g}(\mathbf{x}(t), \mathbf{h}(t)) : \mathbb{R}^u \times \mathbb{R}^v \to \mathbb{R}^v$, with the starting condition $\mathbf{h}(0) = \mathbf{h}_0$ and the solution $\mathbf{h}(t) : \mathbb{R} \to \mathbb{R}^v$: $\frac{d}{dt}\mathbf{h}(t) = \mathbf{g}(\mathbf{h}(t), \mathbf{x}(t))$, such that $F_{\mathbf{g}}(\mathbf{x}(\cdot), \mathbf{h}_0; t) = \mathbf{h}(t)$. Not all functions on trajectories can be represented this way.

**Start-corrected differentially-definable mappings,**    i.e. the elements of $\mathcal{F}$, minus their initial state $\mathbf{h}_0$: $\mathring{\mathcal{F}} = \{\mathring{F}_{\mathbf{g}}(\mathbf{x}(\cdot), \mathbf{h}_0; t) = F_{\mathbf{g}}(\mathbf{x}(\cdot), \mathbf{h}_0; t) - \mathbf{h}_0 | F_{\mathbf{g}} \in \mathcal{F}\}$. Since we set $\mathbf{h}_0 = 0$ in most practical cases, the distinction is rather technical; it is here mostly to account for Lipschitzness w.r.t. $\mathbf{h}$, as we will see below.

**Lipschitz constraints.**    We introduce two ways to constrain this class with Lipschitz constants: with $\mathbf{g}$ being $M_x, M_h$-Lipschitz

$$\mathring{\mathcal{F}}_g(M_x, M_h) = \{\mathring{F}_{\mathbf{g}} \in \mathring{\mathcal{F}} | \mathbf{g} - M_x, M_h\text{-lipschitz w.r.t. } \mathbf{x}, \mathbf{h}\},$$

and with $\mathring{F}_{\mathbf{g}}(\ldots; t)$ being $L_x(t), L_h(t)$-lipschitz, where $L_x(t), L_h(t) : \mathbb{R} \to \mathbb{R}_+$:

$$\mathring{\mathcal{F}}_F(L_x(\cdot), L_h(\cdot)) = \{\mathring{F}_{\mathbf{g}} \in \mathring{\mathcal{F}} | \mathring{F}_{\mathbf{g}} - L_x(t), L_h(t)\text{-lipschitz w.r.t. } \mathbf{x}(\cdot), \mathbf{h}_0\},$$

where Lipschitzness w.r.t. $\mathbf{x}(\cdot)$ is meant in terms of $L_2$ function norms: $\|\mathbf{x}(\cdot)\|_{L_2} = \sqrt{\int \|\mathbf{x}(t)\|_2^2 dt}$. Intuitively, $\mathring{\mathcal{F}}_g(M_x, M_h)$ represents all mappings that our NCDE can learn, given a Lipschitz-constrained Neural Network $\mathbf{g}$, while $\mathring{\mathcal{F}}_F(L_x(\cdot), L_h(\cdot))$ represents all the possible Lipschitz-mappings that we may want to learn.

Equipped with the above definitions, we can formulate the following:

**Theorem** (3.1). *The classes $\mathring{\mathcal{F}}_g(M_x, M_h)$ and $\mathring{\mathcal{F}}_F(L_x(\cdot), L_h(\cdot))$ are equal, given the following relations between their arguments:*

$$L_x(t) = M_x \sqrt{\frac{1}{2M_h}\left(e^{2M_h t} - 1\right)}; L_h(t) = e^{M_h t} - 1.$$

*Proof.* We conduct the proof in two steps: first proving the subset relation $\mathring{\mathcal{F}}_g(M_x, M_h) \subseteq \mathring{\mathcal{F}}_F(L_x(\cdot), L_h(\cdot))$, and then the converse $\mathring{\mathcal{F}}_F(L_x(\cdot), L_h(\cdot)) \subseteq \mathring{\mathcal{F}}_g(M_x, M_h)$. Together, these relations will imply equality of the two considered classes.

**Subset relation.**    First, we prove the relation $\mathring{\mathcal{F}}_g(M_x, M_h) \subseteq \mathring{\mathcal{F}}_F(L_x(\cdot), L_h(\cdot))$. Let's consider an $M_x, M_h$-Lipschitz $\mathbf{g}$, and prove that the corresponding $\mathring{F}$ is $L_x, L_h$-Lipschitz.

Say, we have two trajectories, $\mathbf{x}_1(t), \mathbf{x}_2(t)$, each producing their repsective $\mathbf{h}_{1,2}(t)$, with $\mathbf{h}_1(0) \neq \mathbf{h}_2(0)$. The distance between the corresponding $\mathring{F}$-s will be given by:

$$\|(F(\mathbf{x}_1, \mathbf{h}_0, t) - F(\mathbf{x}_1, \mathbf{h}_0, 0)) - (F(\mathbf{x}_2, \mathbf{h}_0, t) - F(\mathbf{x}_2, \mathbf{h}_0, 0))\| \triangleq \|\Delta(t)\|,$$

where we denote the expression under the norm by $\Delta$ for brevity. The absolute derivative of the norm is always less or equal to the norm of the derivative, which can be proven using the Cauchy-Schwartz inequality:

$$\frac{d}{dt}\|\Delta\|_2 \leq \left|\frac{d}{dt}\|\Delta\|_2\right| = \left|\frac{d}{dt}\sqrt{\Delta \cdot \Delta}\right| = \frac{\Delta \cdot \Delta'}{\|\Delta\|_2} \leq \frac{\|\Delta\|_2 \cdot \|\Delta'\|_2}{\|\Delta\|_2} = \|\Delta'\|_2 =$$

Here $\Delta'$ is the derivative of $\Delta$ w.r.t. $t$. We can write it using $\mathbf{g}$, the starting points cancel out:

$$= \|\Delta'(t)\|_2 = \|\mathbf{g}(\mathbf{x}_1(t), \mathbf{h}_1(t)) - \mathbf{g}(\mathbf{x}_2(t), \mathbf{h}_2(t))\|_2 =$$

We add and subtract a "mixed" term $\mathbf{g}(\mathbf{x}_1(t), \mathbf{h}_2(t))$:

$$= \|\mathbf{g}(\mathbf{x}_1(t), \mathbf{h}_1(t)) - \mathbf{g}(\mathbf{x}_1(t), \mathbf{h}_2(t)) + \mathbf{g}(\mathbf{x}_1(t), \mathbf{h}_2(t)) - \mathbf{g}(\mathbf{x}_2(t), \mathbf{h}_2(t))\|_2 \leq$$

The norm of that sum may be bounded using the triangle rule:

$$\leq \|\mathbf{g}(\mathbf{x}_1(t), \mathbf{h}_1(t)) - \mathbf{g}(\mathbf{x}_1(t), \mathbf{h}_2(t))\|_2 + \|\mathbf{g}(\mathbf{x}_1(t), \mathbf{h}_2(t)) - \mathbf{g}(\mathbf{x}_2(t), \mathbf{h}_2(t))\|_2 \leq$$

And both of these two norm-terms can now be bounded by the Lipschitz property:

$$\leq M_x\|\mathbf{x}_1 - \mathbf{x}_2\|_2 + M_h\|\mathbf{h}_1 - \mathbf{h}_2\|_2 =$$

We add and subtract $\mathbf{h}_1(0) - \mathbf{h}_2(0) \triangleq \Delta_0$ in the second term:

$$M_x\|\mathbf{x}_1 - \mathbf{x}_2\|_2 + M_h\|\mathbf{h}_1(t) - \mathbf{h}_2(t) - (\mathbf{h}_1(0) - \mathbf{h}_2(0)) + (\mathbf{h}_1(0) - \mathbf{h}_2(0))\|_2 \leq$$

We now notice that we can substitute $\Delta$ in the second term, and upper-bound it with the triangle inequality

$$\leq M_x \|\mathbf{x}_1 - \mathbf{x}_2\|_2 + M_h \|\Delta\|_2 + M_h \|\Delta_0\|_2,$$

where we denote $\Delta_0 \triangleq \mathbf{h}_1(0) - \mathbf{h}_2(0)$.

Putting the above reasoning together, we are left with the differential inequality:

$$\frac{d}{dt}\|\Delta\| \leq M_x \|\mathbf{x}_1 - \mathbf{x}_2\| + M_h \|\Delta\| + M_h \|\Delta_0\|.$$

Using the standard ODE trick, we assume that $\Delta(t) = C(t)e^{M_h t}$, and substitute this into the inequality:

$$\frac{d}{dt}\|\Delta\| = \frac{dC}{dt}e^{M_h t} + C(t)M_h e^{M_h t} \leq M_x \|\mathbf{x}_1 - \mathbf{x}_2\| + M_h C(t)e^{M_h t} + M_h \|\Delta_0\|.$$

The terms $M_h C(t)e^{M_h t}$ cancel out:

$$\frac{dC}{dt}e^{M_h t} \leq M_x \|\mathbf{x}_1 - \mathbf{x}_2\| + M_h \|\Delta_0\|.$$

Now, we separate the variables, and integrate from $\tau = 0$ to $t$ :

$$\int_{\tau=0}^{t} dC = C(t) - C(0) \leq \left( M_x \int_{\tau=0}^{t} \|\mathbf{x}_1(\tau) - \mathbf{x}_2(\tau)\|e^{-M_h \tau} d\tau \right) +$$

$$+ M_h \|\Delta_0\| \int_{\tau=0}^{t} e^{-M_h \tau} d\tau.$$

The integral without $\mathbf{x}_{1,2}(t)$ can be integrated analytically:

$$\int_{\tau=0}^{t} e^{-M_h \tau} d\tau = \frac{1}{M_h} \left( 1 - e^{-M_h t} \right).$$

The integral with $\mathbf{x}_{1,2}$ can be bounded using Cauchy-Schwartz:

$$\int_{\tau=0}^{t} \|\mathbf{x}_1(\tau) - \mathbf{x}_2(\tau)\|e^{-M_h \tau} d\tau \leq$$

$$\leq \sqrt{\int_{\tau=0}^{t} \|\mathbf{x}_1(\tau) - \mathbf{x}_2(\tau)\|^2 d\tau} \sqrt{\int_{\tau=0}^{t} e^{-2M_h \tau} d\tau}$$

The first term is the $L_2$-distance between the two input signals, we denote it as $\rho(\mathbf{x}_1, \mathbf{x}_2)$; the second one can be integrated analytically:

$$\int_{\tau=0}^{t} e^{-2M_h \tau} d\tau = \frac{1}{2M_h} \left( 1 - e^{-2M_h t} \right).$$

As a starting condition, we know that $\Delta(0) = 0$ (since it is start-corrected), so $C(0) = 0$. Now we have an inequality for $C(t)$:

$$C(t) \leq \|\Delta_0\|_2 \left( 1 - e^{-M_h t} \right) + M_x \rho(\mathbf{x}_1, \mathbf{x}_2) \sqrt{\frac{1}{2M_h} \left( 1 - e^{-2M_h t} \right)}.$$

Multiplying it by $e^{M_h t}$ gives us the inequality for $\Delta(t)$:

$$\|\Delta(t)\| \leq \|\Delta_0\|_2 \left( e^{M_h t} - 1 \right) + M_x \rho(\mathbf{x}_1, \mathbf{x}_2) \sqrt{\frac{1}{2M_h} \left( e^{2M_h t} - 1 \right)}.$$

To get the individual expression for $L_x$, we simply set $\Delta_0$ to zero. To get the expression for $L_h$, we set $\rho$ to zero.

**Superset relation.** Now we prove the converse relationship: $\mathring{\mathcal{F}}_F(L_x(\cdot), L_h(\cdot)) \subseteq \mathring{\mathcal{F}}_g(M_x, M_h)$.

Assume that we have $F \in \mathring{\mathcal{F}}_F(L_x(\cdot), L_h(\cdot))$, where for some $M_x, M_h$:

$$L_x(t) = M_x\sqrt{\frac{1}{2M_h}\left(e^{2M_ht} - 1\right)}; L_h(t) = e^{M_ht} - 1.$$

We need to prove that the corresponding $\mathbf{g}$ is $M_x, M_h$-Lipschitz.

**We start with the $x$ component.** Consider two input trajectories $\mathbf{x}_1 \neq \mathbf{x}_2$, and equal starting points $\mathbf{h}_1 = \mathbf{h}_2 = \mathbf{h}_0$. Since $F$ is Lipschitz, we can write the following inequality:

$$\|\mathring{F}(\mathbf{x}_1, \mathbf{h}_0) - \mathring{F}(\mathbf{x}_2, \mathbf{h}_0)\| \triangleq \|\Delta\| \leq \rho(\mathbf{x}_1, \mathbf{x}_2)M_x\sqrt{\frac{1}{2M_h}\left(e^{2M_ht} - 1\right)}$$

Here, we will also denote the difference between these trajectories by $\Delta$:

$$\mathring{F}(\mathbf{x}_1, \mathbf{h}_0) - \mathring{F}(\mathbf{x}_2, \mathbf{h}_0) = \Delta.$$

We square the Lipschitz inequality, and take the derivative w.r.t. $t$:

$$2(\Delta\mathbf{g}, \Delta) \leq \frac{M_x^2}{2M_h}\left(\frac{d\rho^2}{dt}\left(e^{2M_ht} - 1\right) + \rho^2\frac{d}{dt}\left(e^{2M_ht} - 1\right)\right).$$

Here, we use the notation $\Delta\mathbf{g} = \mathbf{g}(\mathbf{x}_1(t), \mathbf{h}_1(t)) - \mathbf{g}(\mathbf{x}_2(t), \mathbf{h}_2(t))$. The derivatives of $\rho^2$ and $\left(e^{2M_ht} - 1\right)$ are easily calculated analytically:

$$\frac{d\rho^2}{dt} = \|\mathbf{x}_1(t) - \mathbf{x}_2(t)\|^2.$$

$$\frac{d}{dt}\left(e^{2M_ht} - 1\right) = 2M_he^{2M_ht}.$$

We substitute these expressions:

$$2(\Delta\mathbf{g}, \Delta) \leq \frac{M_x^2}{2M_h}\left(\|\mathbf{x}_1(t) - \mathbf{x}_2(t)\|^2 \cdot \left(e^{2M_ht} - 1\right) + \rho^2 2M_he^{2M_ht}\cdot\right).$$

Now we divide the above by $t$, and take the limit at $t \to 0$.

$$\lim_{t\to0}2(\Delta\mathbf{g}, \frac{1}{t}\Delta) \leq \lim_{t\to0}\frac{M_x^2}{2M_h}\left(\|\mathbf{x}_1(t) - \mathbf{x}_2(t)\|^2\frac{1}{t}\left(e^{2M_ht} - 1\right) + \frac{1}{t}\rho^2 2M_he^{2M_ht}\right)$$

We calculate the limits for terms with $\frac{1}{t}$:

$$\lim_{t\to0}\frac{1}{t}\Delta = \Delta\mathbf{g}(0);$$

$$\lim_{t\to0}\frac{1}{t}\left(e^{2M_ht} - 1\right) = 2M_h;$$

$$\lim_{t\to0}\frac{1}{t}\rho^2 = \|\mathbf{x}_1(0) - \mathbf{x}_2(0)\|^2.$$

The limit for the other terms is achieved by simply setting $t = 0$:

$$2(\Delta\mathbf{g}(0), \Delta\mathbf{g}(0)) \leq \frac{M_x^2}{2M_h}\left(\|\mathbf{x}_1(0) - \mathbf{x}_2(0)\|^2 2M_h + \|\mathbf{x}_1(0) - \mathbf{x}_2(0)\|^2 2M_h\right).$$

Simplifying, we get:

$$\|\mathbf{g}(\mathbf{x}_1(0), \mathbf{h}_0) - \mathbf{g}(\mathbf{x}_2(0), \mathbf{h}_0)\|^2 \leq M_x\|\mathbf{x}_1(0) - \mathbf{x}_2(0)\|^2.$$

Since the choice of $\mathbf{x}_1(0), \mathbf{x}_2(0), \mathbf{h}_0$ is arbitrary, the above implies that $\mathbf{g}$ is $M_x$-Lipschitz.

**Now for the $h$ component.** We follow a very similar approach, but this time the input trajectories are the same $\mathbf{x}_1 = \mathbf{x}_2 = \mathbf{x}$, while the starting points are not $\mathbf{h}_1 \neq \mathbf{h}_2$. Therefore, since $F$ is Lipschitz, we can write the following inequality:

$$\|\mathring{F}(\mathbf{x}, \mathbf{h}_1; t) - \mathring{F}(\mathbf{x}, \mathbf{h}_2; t)\| \triangleq \|\Delta(t)\| \leq \|\Delta_0\| \left(e^{M_h t} - 1\right).$$

Here, $\Delta_0 \triangleq \mathbf{h}_1 - \mathbf{h}_2$. Again, we take the square and differentiate w.r.t. $t$:

$$2(\Delta, \Delta \mathbf{g}) \leq \|\Delta_0\|^2 2 \left(e^{M_h t} - 1\right) M_h e^{M_h t}.$$

Divide by $t$ and take the limit at $t \to 0$:

$$2\|\Delta \mathbf{g}\|^2 \leq \|\Delta_0\|^2 2 M_h.$$

Since the choice of $\mathbf{x}, \mathbf{h}_1, \mathbf{h}_2$ is arbitrary, we have proven that $\mathbf{g}$ is $M_h$-Lipschitz.

This concludes the proof of Theorem 3.1. $\qquad\square$

As a corollary, given an $M_x, M_h$-Lipschitz NN $\mathbf{g}$, we can represent all the $L_x, L_h$-Lipschitz (differentially-definable, start-corrected) $F$, and only them.

## B.2 NEGATIVE FEEDBACK MODEL

We model NF by the following simplified differential equation, a specific case within the dynamics (1):

$$\frac{d\mathbf{h}}{dt} = \mathbf{g}_\theta(\hat{\mathbf{x}}(t), \mathbf{h}(t)) = a\mathbf{f}_\theta(\hat{\mathbf{x}}(t), \mathbf{h}(t)) - b\mathbf{h}(t), \ a, b \in \mathbb{R}. \tag{2}$$

The values of $a$ and $b$ determine the relative magnitude of the NF term and the overall scale of the derivative. For clarity, our theoretical analysis focuses on scalar-valued $a$ and $b$, which suffice to demonstrate the key properties. Extending the analysis to vector-valued parameters introduces substantial technical complexity without offering additional insight.

To conduct our analysis, we need to constrain the function $\mathbf{f}_\theta$ and the values $a, b, L_h$:

**Assumption (4.1).** (1) The function $\mathbf{f}_\theta$ from (2) is Lipschitz w.r.t. both $\hat{\mathbf{x}}$ and $\mathbf{h}$ with constants $L_h, L_x$, and (2) it is equal to zero for zero vectors $\mathbf{f}_\theta(\mathbf{0}_u, \mathbf{0}_v) = \mathbf{0}_v$.

**Assumption (4.2).** The values of $a, b, L_h$ are constrained by the following:

$$a \in (0, 1), b \in (0, 1), L_h \in (0, 1), aL_h < b. \tag{3}$$

## B.3 STABILITY

As mentioned in the main text, we approach stability analysis through control systems theory Sontag & Wang (1995). For clarity, we write out the definitions and theorems used in greater detail. First, we introduce the comparison function classes that are used in the stability definition:

**Definition B.1.** Let $\mathcal{K}, \mathcal{L}, \mathcal{KL}$ denote the following classes of functions:

- A continuous function $\gamma : \mathbb{R}_+ \to \mathbb{R}_+$ is said to belong to the comparison class $\mathcal{K}$, if it is increasing with $\gamma(0) = 0$.

- A continuous function $\gamma : \mathbb{R}_+ \to \mathbb{R}_+$ is said to belong to the comparison class $\mathcal{L}$, if it is strictly decreasing with $\gamma(t) \to 0, t \to \infty$.

- A continuous function $\gamma : \mathbb{R}_+ \to \mathbb{R}_+$ is said to belong to the comparison class $\mathcal{K}_\infty$, if it belongs to $\mathcal{K}$ and is unbounded.

- A continuous function $\beta : \mathbb{R}_+^2 \to \mathbb{R}_+$ is said to belong to the comparison class $\mathcal{KL}$, if $\beta(\cdot, t) \in \mathcal{K}, \forall t > 0$, and $\beta(r, \cdot) \in \mathcal{L}, \forall r > 0$.

Now, we can formulate the definition of stability in the control-systems sense:

**Definition B.2.** A system (1) is said to be input-to-state stable (ISS), if there exist $\gamma \in \mathcal{K}$, and $\beta \in \mathcal{KL}$, s.t.:

$$\|\mathbf{h}(t)\|_2 \leq \beta(\|\mathbf{h}_0\|_2, t) + \gamma(\|\hat{\mathbf{x}}\|_\infty). \tag{5}$$

Just like with traditional ODE stability, instead of directly testing the definition, it is often more convenient to construct an *ISS-Lyapunov function*:

**Definition B.3.** A smooth function $V : \mathbb{R}^v \to \mathbb{R}_+$ is called an ISS-Lyapunov function, if there exist functions $\psi_1, \psi_2 \in \mathcal{K}_\infty$ and $\alpha, \chi \in \mathcal{K}$, s.t.

$$\psi_1(\|\mathbf{h}\|_2) \leq V(\mathbf{h}) \leq \psi_2(\|\mathbf{h}\|_2), \forall \mathbf{h} \in \mathbb{R}^v, \tag{13}$$

and

$$\forall \mathbf{h} : \|\mathbf{h}\|_2 \geq \chi(\|\mathbf{x}\|_2) \to \nabla V \cdot \mathbf{g}_\theta \leq -\alpha(\|\mathbf{h}\|_2). \tag{14}$$

The connection between Definitions B.2 and B.3 is given by the following lemma Sontag & Wang (1995):

**Lemma B.4.** *The ODE* (1) *is ISS, if and only if a corresponding ISS-Lyapunov function exists.*

Now, using the facts formulated above, we can proceed to proving Theorem 4.4:

**Theorem** (4.4). *The ODE* (2) *under Assumptions 4.1, 4.2 is input-to-state stable.*

*Proof.* Let us build a corresponding ISS-Lyapunov function. According to Lemma B.4, this proves ISS. Consider $V(\mathbf{h}) \triangleq \frac{1}{2}\|\mathbf{h}\|_2^2$. Equation (13) evidently stands (e.g. $\psi_{1,2}(\|\mathbf{h}\|_2) \equiv \frac{1}{2}\|\mathbf{h}\|_2^2$). The gradient of the squared half-norm is the vector itself: $\nabla V = \mathbf{h}$. So, to test (14), we can bound the corresponding dot-product:

$$(\nabla V, \mathbf{g}_\theta) = (\mathbf{h}, a\mathbf{f}_\theta - b\mathbf{h}) \leq a\|\mathbf{h}\|_2\|\mathbf{f}_\theta\|_2 - b\|\mathbf{h}\|_2^2.$$

Using Assumption 4.1, we can bound $\|\mathbf{f}_\theta\|_2$:

$$\begin{aligned}
\|\mathbf{f}_\theta(\hat{\mathbf{x}}, \mathbf{h})\|_2 &= \|\mathbf{f}_\theta(\hat{\mathbf{x}}, \mathbf{h}) - \mathbf{f}_\theta(\hat{\mathbf{x}}, \mathbf{0}_v) + \mathbf{f}_\theta(\hat{\mathbf{x}}, \mathbf{0}_v) - \mathbf{f}_\theta(\mathbf{0}_u, \mathbf{0}_v) + \mathbf{f}_\theta(\mathbf{0}_u, \mathbf{0}_v)\|_2 \\
&\leq \|\mathbf{f}_\theta(\hat{\mathbf{x}}, \mathbf{h}) - \mathbf{f}_\theta(\hat{\mathbf{x}}, \mathbf{0}_v)\|_2 + \|\mathbf{f}_\theta(\hat{\mathbf{x}}, \mathbf{0}_v) - \mathbf{f}_\theta(\mathbf{0}_u, \mathbf{0}_v)\|_2 + \|\mathbf{f}_\theta(\mathbf{0}_u, \mathbf{0}_v)\|_2 \\
&\leq L_h\|\mathbf{h}\|_2 + L_x\|\hat{\mathbf{x}}\|_2 + 0.
\end{aligned}$$

Now we have:

$$(\nabla V, g_\theta) \leq a\|\mathbf{h}\|_2(L_h\|\mathbf{h}\|_2 + L_x\|\hat{\mathbf{x}}\|_2) - b\|\mathbf{h}\|_2^2 = (aL_h - b)\|\mathbf{h}\|_2^2 + aL_x\|\hat{\mathbf{x}}\|_2\|\mathbf{h}\|_2. \tag{15}$$

The final expression is a quadratic polynomial. The coefficient next to $\|\mathbf{h}\|_2^2$ is negative, according to (3). Consequently, it will decrease for large-enough values of $\|\mathbf{h}\|_2$, which allows us to construct functions $\chi, \alpha$, satisfying (14). For some fixed $\varepsilon \in (0, 1)$, consider the following function:

$$\chi(r) \triangleq \frac{aL_x}{b - aL_h} r(1 - \varepsilon)^{-1}; \; \chi^{-1}(r) = \frac{b - aL_h}{aL_x}(1 - \varepsilon)r.$$

Then, for any $\mathbf{h} : \|\mathbf{h}\|_2 \geq \chi(\|\hat{\mathbf{x}}\|_2)$, which is in our case equivalent to $\|\hat{\mathbf{x}}\|_2 \leq \chi^{-1}(\|\mathbf{h}\|_2)$, we can bound (15):

$$\begin{aligned}
(\nabla V, g_\theta) &\leq (aL_h - b)\|\mathbf{h}\|_2^2 + aL_x\|\hat{\mathbf{x}}\|_2\|\mathbf{h}\|_2 \\
&\leq (aL_h - b)\|\mathbf{h}\|_2^2 + aL_x\frac{b - aL_h}{aL_x}(1 - \varepsilon)\|\mathbf{h}\|_2^2 \\
&= -\varepsilon(b - L_h a)\|\mathbf{h}\|_2^2 \triangleq -\alpha(\|\mathbf{h}\|_2).
\end{aligned}$$

Since $b - L_h a > 0$, the functions we have constructed lie in the correct classes: $\alpha, \chi \in \mathcal{K}$. Consequently, $V$ is a valid ISS-Lyapunov function, which implies that our system is ISS-stable. $\square$

## B.4 ERROR BOUNDS

To devise the theoretical results of this section, we need to introduce the "perfect" Neural CDE, calculated without any errors:

$$\mathbf{h}^*(0) = \mathbf{h}_0; \quad \frac{d\mathbf{h}^*(t)}{dt} = a\mathbf{f}_\theta(\mathbf{x}(t), \mathbf{h}^*(t)) - b\mathbf{h}^*(t). \tag{6}$$

Then, both the integration and the interpolation errors may be modelled via the following "imperfect" ODE, often referred to as the "Modified Vector Field" Reich (1999):

$$\mathbf{h}(0) = \mathbf{h}_0; \quad \frac{d\mathbf{h}(t)}{dt} = a\mathbf{f}_\theta(\mathbf{x}(t), \mathbf{h}(t)) - b\mathbf{h}(t) + \xi(t), \tag{7}$$

where $\xi(t)$ is a random vector, which represents the momentary deviation of our numerical ODE from the true one. For further theoretical considerations, we introduce two types of bounds on $\xi(t)$:

$$\text{Pointwise: } \mathbb{E}\,\|\xi(t)\|_2^2 \le \xi_{\text{PW}}^2, \forall t; \quad \text{Interval: } \frac{1}{t_b - t_a} \int_{t_a}^{t_b} \mathbb{E}\,\|\xi(t)\|_2^2 \, dt \le \xi_{\text{INT}}^2, \forall t_a < t_b. \tag{8}$$

For each of the introduced bounds, we can devise an estimate for the error in the final hidden state:

**Theorem** (**4.5**). *Error Bounds (Robustness). Consider the differential equations* (6),(7), *under Assumptions 4.1, 4.2 Then, depending on whether we are working with the pointwise or the interval errors* (8), *we can write different bounds on the variance, accumulating in* **h**.

Pointwise bound:
$$\mathbb{E}\|\mathbf{h}(T) - \mathbf{h}^*(T)\|_2^2 \le \left(\frac{a}{b - aL_h}\right)^2 \xi_{\text{PW}}^2. \tag{9}$$

Interval bound:
$$\frac{1}{T} \int_{t_a}^{t_b} \mathbb{E}\,\|\mathbf{h}(t) - \mathbf{h}^*(t)\|_2^2 \, dt \le \left(\frac{a}{b - aL_h}\right)^2 \xi_{\text{INT}}^2. \tag{10}$$

*Proof.* Consider the time-derivative of the squared error norm, multiplied by $\frac{1}{2}$ for convenience:

$$\begin{aligned}
\frac{1}{2}\frac{d}{dt}\|\mathbf{h}(t) - \mathbf{h}^*(t)\|_2^2 &= \left(\frac{d(\mathbf{h}(t) - \mathbf{h}^*(t))}{dt}, \mathbf{h}(t) - \mathbf{h}^*(t)\right) = \\
&= (a\delta_f - b\delta_h + \xi, \delta_h) = \\
&= a(\delta_f + \xi, \delta_h) - b\|\delta_h\|_2^2 = \\
&\le a(L_h\,\|\delta_h\|_2^2 + \|\xi\|\,\|\delta_h\|_2) - b\,\|\delta_h\|_2^2 = \\
&= (aL_h - b)\,\|\delta_h\|_2^2 + a\,\|\xi\|_2\,\|\delta_h\|_2\,, \tag{16}
\end{aligned}$$

where we introduce the following notation for conciseness:

$$\begin{aligned}
\delta_f &\triangleq \mathbf{f}_\theta(\mathbf{x}, \mathbf{h}) - \mathbf{f}_\theta(\mathbf{x}, \mathbf{h}^*) \\
\delta_h &\triangleq \mathbf{h}(t) - \mathbf{h}^*(t), \\
\delta_x &\triangleq \hat{\mathbf{x}}(t) - \mathbf{x}(t).
\end{aligned}$$

If we have a pointwise bound on $\mathbb{E}\,\|\xi\|_2^2$, the expression from (16) can be upper bounded by a quadratic function, where the multiplier $aL_h - b$ next to the squared term is negative, since $aL_h < b$. It turns negative for larger $\|\delta_h\|_2$:

$$\text{if} \quad \|\delta_h\|_2 > \frac{a\xi_{\text{PW}}}{b - aL_h}, \quad \text{then} \quad (aL_h - b)\,\|\delta_h\|_2^2 + \xi_{\text{PW}}\,\|\delta_h\|_2 < 0$$

$$\implies \frac{1}{2}\mathbb{E}\frac{d}{dt}\,\|\mathbf{h}(t) - \mathbf{h}^*(t)\|_2^2 \le 0,$$

which proves half of this theorem.

However, if we only have an interval error bound, we are forced to integrate (16) on the interval $[0, T]$:

$$\frac{1}{2}\int_0^T \frac{d}{dt}\,\|\mathbf{h}(t) - \mathbf{h}^*(t)\|_2^2 \, dt \le \int_0^T \|\delta_h(t)\|_2^2 \,(aL_h - b)dt + \int_0^T \|\xi(t)\|_2\,\|\delta_h(t)\|_2 \, dt. \tag{17}$$

Let us rewrite the last term, dropping all constants, and use the Cauchy–Schwarz inequality for integrals:

$$\int_0^T \|\xi(t)\|_2\,\|\delta_h(t)\|_2 \, dt \le \sqrt{\int_0^T \|\xi(t)\|_2^2 \, dt \int_0^T \|\delta_h(t)\|_2^2 \, dt} \triangleq RH,$$

where we introduced the following notation:

$$R^2 \triangleq \int_0^T \|\xi(t)\|_2^2 \, dt, \ \ H^2 \triangleq \int_0^T \|\delta_h(t)\|_2^2 \, dt.$$

In light of that, let's rewrite (17):

$$\frac{1}{2} \left( \|\delta_h(T)\|_2^2 - \|\delta_h(0)\|_2^2 \right) = \frac{1}{2} \|\delta_h(T)\|_2^2 \le (aL_h - b)H^2 + aRH\sqrt{k},$$

where we used $\delta_h(0) = 0$. The right-hand side is a quadratic function with respect to $H$; again, the multiplier next to the squared term is negative. Simultaneously, the left-hand side is positive, being a norm. Together, this means that $H$ needs to be in between the roots of this quadratic function to satisfy the positivity constraints:

$$0 \le H \le \frac{aR\sqrt{k}}{b - aL_h}.$$

We square the above, and then substitute $R, H$ in:

$$\int_0^T \|\delta_h(t)\|_2^2 \le \left( \frac{a}{b - aL_h} \right)^2 \int_0^T \|\xi(t)\|_2^2 \, dt.$$

Dividing by $T$ and taking the expectation (which we may move under the integral due to linearity), we get:

$$\frac{1}{T} \int_0^T \mathbb{E} \|\delta_h(t)\|_2^2 \le \left( \frac{a}{b - aL_h} \right)^2 \frac{1}{T} \int_0^T \mathbb{E} \|\xi(t)\|_2^2 \, dt \le \left( \frac{a}{b - aL_h} \right)^2 \xi_{\text{INT}}.$$

$\square$

### B.4.1 BOUND TIGHTNESS

To analyse the tightness of bounds from Theorem 4.5, we provide specific examples, which adhere to our assumptions while achieving the formulated bounds.

**Example 1.** To simplify analysis, assume $u = v = 1$; a similar example may be provided for the multidimensional case. Consider the following scenario:

$$\frac{dh}{dt} = A + (1 - \varepsilon/2)h - (1 + \varepsilon/2)h, \ h(0) = 0; \tag{18}$$

$$\hat{x}(t) \equiv A; a = 1; b = 1 + \varepsilon/2; L_h = 1 - \varepsilon/2; L_x = 1. \tag{19}$$

In this case, $b - aL_h = \varepsilon > 0$. The solution is:

$$h(t) = \frac{-Ae^{-\varepsilon t} + A}{\varepsilon}.$$

On the other hand, suppose the true value of $x$ is also constant, $x(t) \equiv B \in \mathbb{R}$. Then, the corresponding "true" solution is given by:

$$h^*(t) = \frac{-Be^{-\varepsilon t} + B}{\varepsilon}.$$

Then, the errors (9), (10) from Theorem 4.5 can be calculated analytically:

$$\mathbb{E}[h(t) - h^*(t)]^2 = \left( \frac{(A - B)(1 - e^{-\varepsilon t})}{\varepsilon} \right)^2 \to \frac{(A - B)^2}{\varepsilon^2}, \ t \to \infty;$$

$$\int_{t_1}^{t_2} \mathbb{E}[h(t) - h^*(t)]^2 dt = \int_{t_1}^{t_2} \left( \frac{(A - B)(1 - e^{-\varepsilon t})}{\varepsilon} \right)^2 dt \to \frac{1}{\varepsilon^2}(t_2 - t_1)(A - B)^2, \ t_1 \to \infty.$$

They match the corresponding bounds. Note, that to illustrate the asymptotic of the interval bound, we consider $t_1 > 0$, and calculate limit of the interval bound at $t_1 \to \infty$ (instead of integrating on $[0, T]$). This allows us to show how the bound behaves on large values of $t$.

## B.5 GP INTERPOLATION ERROR

Now, let's devise the Gaussian Process interval-error estimates. Let $\mathbf{x}(t)$ be a stationary zero-mean $u$-dimensional Gaussian process (GP) with the covariance function $\mathbf{K}(\cdot, \cdot) = \{K_i(\cdot, \cdot)\}$ and independent components:

$$
\begin{aligned}
\mathbb{E}\mathbf{x}(t) &= \mathbf{0}, \\
\mathbb{E}\left[\mathbf{x}(t_1) \odot \mathbf{x}(t_2)\right] &= \mathbf{K}(t_1, t_2), \\
\mathbb{E}\left[x_i(t) \cdot x_j(t)\right] &= 0 \text{ for } i \neq j, \\
\mathbf{K}(t_1, t_2) &= \mathbf{K}(t_2 - t_1),
\end{aligned}
\tag{20}
$$

where the symbol $\odot$ denotes component-wise multiplication, and lower indexing $x_i$ denotes vector components.

Suppose we know the values of (20) at timestamps $0 = t_1 < t_2 \ldots < t_n = T \in \mathbb{R}$. Consider the corresponding maximum a posteriori predictor $\hat{\mathbf{x}}(t)$:

$$
\hat{\mathbf{x}}(t) = \operatorname*{argmax}_{\mathbf{x}(t)} p\left[\mathbf{x}(t) | \mathbf{x}(t_1) = \mathbf{x}_1, \ldots, \mathbf{x}(t_n) = \mathbf{x}_n\right].
$$

We seek to devise two types of uniform error bounds for the predictor:

$$
\sigma_{\text{PW}}^2 \triangleq \mathbb{E}\left\|\hat{\mathbf{x}}(t) - \mathbf{x}(t)\right\|_2^2 \leq \bar{\sigma}_{\text{PW}}^2, \qquad \sigma_{\text{INT}}^2 \triangleq \int_{t_k}^{t_{k+1}} \mathbb{E}\left\|\hat{\mathbf{x}}(t) - \mathbf{x}(t)\right\|_2^2 dt \leq \bar{\sigma}_{\text{INT}}^2, \tag{21}
$$

where $\bar{\sigma}_{\text{INT}}, \bar{\sigma}_{\text{PW}}$ are the pointwise and the interval error bounds, constants independent of $t$ and $k$. The pointwise expression is more convenient; however, the interval one is often tighter.

### B.5.1 GP ASSUMPTIONS

To simplify the analysis, we make two assumptions about the nature of the sequences we are working with. However, they do not limit the applicability of our results.

Most prior theoretical results are devised for the case of uniform sequences. To extend the bounds towards irregular data, we assume that changing interval sizes predictably changes the resulting error:

**Assumption B.5.** Let $S = \{(t_k, \mathbf{x}_k)\}_k$ be a given irregular time series. We construct an alternative time series $\tilde{S} = \{(\tilde{t}_k, \mathbf{x}_k)\}_k$, with an additional (possibly negative) margin $r$, such that $t_m - t_{m+1} < r < +\infty$, introduced between the $m$-th and $m+1$-th points:

$$
\forall k \quad \begin{cases} \tilde{t}_k = t_k, k \leq m; \\ \tilde{t}_k = t_k + r, k > m. \end{cases}
$$

Consider the pointwise errors (21) for the original $S$ and modified $\tilde{S}$ sequences: $\sigma^2$ and $\tilde{\sigma}^2$, respectively. We assume they relate as follows:

$$
\forall t \in \mathbb{R} \quad \begin{cases} \sigma^2(t) \leq \tilde{\sigma}^2(\tilde{t}) & \text{for } r > 0, \\ \sigma^2(t) \geq \tilde{\sigma}^2(\tilde{t}) & \text{for } r < 0, \end{cases}
$$

where $\tilde{t} = t + \mathbb{I}[t > t_{m+1}]r$, and $\mathbb{I}$ denotes the indicator function.

The respective covariance determines the amount of information each observation provides about the unknown value. If all covariances decrease, the variance of $\hat{\mathbf{x}}$ will increase and vice versa. We have also empirically tested this assumption in Appendix B.5.4, and consider a specific theoretical setting where this assumption holds in Appendix B.5.3.

Assumption B.5 is formulated for pointwise errors, however similar conclusions for interval errors follow as corollaries:

**Corollary B.6.** *Under the notation of Assumption B.5, the corresponding interval errors relate similarly:*

$$
\forall k \quad \begin{cases} \sigma_k^2 \leq \tilde{\sigma}_k^2 & \text{for } r > 0, \\ \sigma_k^2 \geq \tilde{\sigma}_k^2 & \text{for } r < 0. \end{cases}
$$

Finally, we can extend all the devised error bounds to irregular sequences.

**Corollary B.7.** *Consider an irregular time series $S = \{(t_k, \mathbf{x}_k)\}$. Let $\delta_{\min}, \delta_{\max}$ be the minimum and maximum intervals between observations of S, respectively:*

$$\delta_{\min} = \min_k(t_k - t_{k-1}),$$

$$\delta_{\max} = \max_k(t_k - t_{k-1}).$$

*Consider also two regular series, with $\delta_{\min}, \delta_{\max}$ as the inter-observation intervals:*

$$S_{\min} = \{(k\delta_{\min}, \mathbf{x}_k)\}_k,$$
$$S_{\max} = \{(k\delta_{\max}, \mathbf{x}_k)\}_k.$$

*Then, under the assumption B.5, their interval errors (21) relate as follows:*

$$\sigma^2_{\min,INT} \le \sigma^2_k \le \sigma^2_{\max,INT} \ \forall k.$$

We also make another common assumption:

**Assumption B.8.** The input sequence is infinite:

$$k = -\infty, \ldots, -2, -1, 0, 1, 2, \ldots, +\infty.$$

Prior work Zaytsev & Burnaev (2017) demonstrated that the results for infinite sequences empirically hold for finite ones. Indeed, since a typical kernel decays with an exponential speed, distant observations negligibly impact the resulting error.

### B.5.2 INTERPOLATION ERROR BOUNDS

In this section, we will work under the uniform-grid assumption: according to Corrolary B.7, all the results devised under this assumption can be extended to irregular sequences:

**Assumption B.9.** The sequence $S$ is uniform:

$$t_{k+1} - t_k = \delta, \ \forall k.$$

The optimal estimator for a uniform infinite grid is well-known Kolmogorov (1941):

$$\hat{\mathbf{x}}(t) = \delta \sum_{k=-\infty}^{\infty} \mathbf{K}(t - k\delta) \odot \mathbf{x}(k\delta), \tag{22}$$

where optimal $\mathbf{K}$ is a symmetric kernel that depends on a spectral density. This form allows us to deduce analytical error bounds. For this purpose, we generalise the results from Zaytsev & Burnaev (2017) to multiple dimensions:

**Lemma B.10.** *Let $\mathbf{F}(\omega)$ denote the spectral density of Gaussian process from (20):*

$$\mathbf{F}(\omega) = \int_{-\infty}^{\infty} \exp(2\pi i\omega t)\mathbf{K}(t)dt.$$

*In this notation, under AssumptionsB.8, B.9, and if all components of $\mathbf{K}$ are equal, the following holds:*

$$\mathbb{E}\|\hat{\mathbf{x}}(t) - \mathbf{x}(t)\|^2 = u \int_{-\infty}^{\infty} F(\omega) \left| 1 - \sum_{k \ne 0} e^{2i\pi\omega(t-t_k)} K(t - t_k) \right|^2 d\omega,$$

*where $K(t) \equiv K_i(t)$, and $F(\omega) \equiv F_i(\omega)$ are the components of the kernel function and the spectral density, respectively.*

*Proof.* Let us start with expanding the square:

$$\mathbb{E}\|\hat{\mathbf{x}}(t) - \mathbf{x}(t)\|^2 = \mathbb{E}[\hat{\mathbf{x}}^T(t)\hat{\mathbf{x}}(t)] - 2\mathbb{E}[\hat{\mathbf{x}}^T(t)\mathbf{x}(t)] + \mathbb{E}[\mathbf{x}^T(t)\mathbf{x}(t)].$$

Now, we consider the terms one by one, substituting (22) and rewriting in terms of spectral density. The first term gives:

$$\mathbb{E}[\hat{\mathbf{x}}^T(t)\hat{\mathbf{x}}(t)] = h^2 u \sum_{k,l=-\infty}^{\infty} K(t-t_k)K(t-t_l)K(t_k-t_l) =$$

$$= \int_{-\infty}^{\infty} F(\omega) \left( h^2 u \sum_{k,l=-\infty}^{\infty} K(t-t_k)K(t-t_l)e^{2\pi i\omega(t_l-t_k)} \right) d\omega. \tag{23}$$

The second one:

$$2\mathbb{E}[\hat{\mathbf{x}}^T(t)\mathbf{x}(t)] = 2hu \sum_{k=-\infty}^{\infty} K^2(t-t_k) =$$

$$= \int_{-\infty}^{\infty} F(\omega) \left( 2hu \sum_{k=-\infty}^{\infty} K(t-t_k)e^{2\pi i\omega(t-t_k)} \right) d\omega. \tag{24}$$

And, finally, the third one:

$$\mathbb{E}[\mathbf{x}^T(t)\mathbf{x}(t)] = K(0)u = u \int_{-\infty}^{\infty} F(\omega)\, d\omega. \tag{25}$$

After factoring out the spectral density integral, the terms (23), 24 and (25) are the expansion of a binomial:

$$\left| 1 - h \sum_{k=-\infty}^{\infty} K(t-t_k)e^{2\pi i\omega(t-t_k)} \right|^2 = 1^2 - 2h \sum_{k=-\infty}^{\infty} K(t-t_k)e^{2\pi i\omega(t-t_k)} +$$

$$+ \left( h \sum_{k=-\infty}^{\infty} K(t-t_k)e^{2\pi i\omega(t-t_k)} \right)^2$$

$$\square$$

With Lemma B.10, all other results can be taken directly from Zaytsev & Burnaev (2017), taking into account that $\mathbf{x}(t)$ is $u$-dimensional. We recount select findings from this paper: the general error form, an analytic error expression for a specific kernel, and the minimax error bound.

**Theorem B.11.** *Under the assumptions from Lemma B.10, the error $\sigma_{INT}^2$ from (21), may be written in the following form:*

$$\sigma_{INT}^2 = u\delta \int_{-\infty}^{\infty} F(\omega) \left( (1-\hat{K}(\omega))^2 + \sum_{k=-\infty}^{\infty} \hat{K}\left(\omega + \frac{k}{\delta}\right) \right) d\omega \tag{26}$$

$$= u\delta \int_{-\infty}^{\infty} F(\omega) \frac{\sum_{k\neq 0} F(\omega + \frac{k}{\delta})}{\sum_{k=-\infty}^{\infty} F(\omega + \frac{k}{\delta})} d\omega, \tag{27}$$

*where $\hat{K}$ is the Fourier transform of $K$.*

The error (26) does not depend on the interval number. This is due to the inherent symmetry of an infinite uniform grid: all intervals are identical from the perspective of the covariance function.

Given (26), we can calculate this error for our scenario of cubic spline interpolation. Consider the covariance functions (for some $\theta > 0, Q > 0$):

$$\forall\, i = 1, \ldots, u \colon K_i(t) = \frac{Q}{\omega^4 + \theta^4}. \tag{28}$$

At the limit $\theta \to 0$, the mean of the resulting Gaussian Processes tends to a natural cubic spline Golubev & Krymova (2013). We estimate the corresponding interval error $\sigma_{INT}^2(\theta)$ at $\theta \to 0$, which can be interpreted as the interval error for natural cubic spline interpolation:

**Lemma** (4.6). *Under Assumptions B.5,B.8, for the covariance function* (28) *the interval error* $\sigma^2_{INT}(\theta)$ *from* (21) *has the following asymptotic:*

$$\lim_{\theta \to 0} \sigma^2_{INT}(\theta) = \frac{4\pi^4\sqrt{3}}{63}\delta^4 Q u.$$

*Proof.* We will analytically calculate (26) using techniques from calculus and complex analysis:

$$I(\xi, \delta) \triangleq \int_{-\infty}^{\infty} F(\omega)\frac{\sum_{k \neq 0} F(\omega + \frac{k}{\delta})}{\sum_{k=-\infty}^{\infty} F(\omega + \frac{k}{\delta})}d\omega \triangleq \int_{-\infty}^{\infty} g(\omega, \xi, \delta)d\omega, \tag{29}$$

where the spectral density $F$ is given by 28. Specifically, for splines, we are interested in the following limit:

$$\sigma^2_S \triangleq u\delta \lim_{\xi \to 0} I(\xi, \delta) =? \tag{30}$$

First, we will calculate the infinite sums. After substituting (28) into (29), the integrand will contain:

$$\sum_k \frac{Q}{(\omega + \frac{k}{\delta})^4 + \xi^4} = \dots \tag{31}$$

To calculate this, we devise a general expression for the sum of 4th-degree shifted reciprocals:

$$\sum_k \frac{1}{(k+a)^4 + b^4} = \dots,$$

for some $a, b$, using complex analysis. Consider the following function:

$$f_{R4}(z) = \frac{\pi \cot(\pi z)}{(z+a)^4 + b^4}.$$

It's poles are $z_d = -a + \frac{b}{\sqrt{2}}(\pm 1 \pm i)$ from the denominator, and $z \in \mathbb{Z}$ from the cotangent; all poles are simple ones, and can be calculated using the l'Hopital rule:

$$\text{Res}(f_{R4}, z_0) = \frac{A_{R4}(z_0)}{B'_{R4}(z_0)},$$

where $f_{R4} = A_{R4}/B_{R4}$; $F_{R4}, G'_{R4}$ are analytic at $z_0$. Applying this formula:

$$\text{Res}(f_{R4}, z_d) = \frac{\pi \cot(\pi z_d)}{4(z_d + a)^3},$$

$$\text{Res}(f_{R4}, k) = \frac{1}{(k+a)^4 + b^4}$$

According to the Residuals Theorem, the sum of all poles is zero, so:

$$\sum_k \frac{1}{(k+a)^4 + b^4} = -\sum_{z_d} \frac{\pi \cot(\pi z_d)}{4(z_d + a)^3} \triangleq S_4(a, b).$$

The specific value of $S_4$ is rather unsightly, so we leave it out. Applying this to our case (31):

$$\sum_k \frac{Q}{(\omega + \frac{k}{\delta})^4 + \xi^4} = Q\delta^4 \sum_k \frac{1}{(\omega\delta + k)^4 + (\xi\delta)^4} = Q\delta^4 S_4(\omega\delta, \xi\delta).$$

The integrand from (29) can be re-written as:

$$g = F(\omega)\left(1 - \frac{F(\omega)}{S_4(\omega\delta, \xi\delta)Q\delta^4}\right).$$

Its limit at $\xi \to 0$ is bounded for all values of $\omega$:

$$\lim_{\xi \to 0} g(\xi, \omega, \delta) = Q\delta^4\pi^4\frac{-2z^4 \sin^2(z) + 3z^4 - 3\sin^4(z)}{z^8(\cos(2z) + 2)},$$

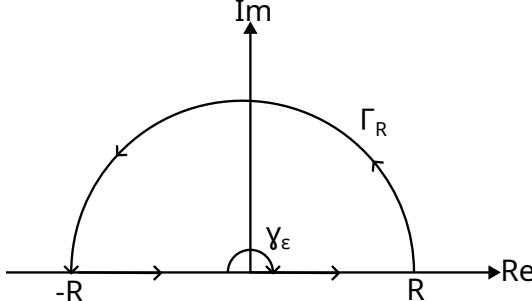

Figure 5: Semicircle contour

where we denote $z = \pi\delta\omega$. This value has a finite limit at $\omega = 0$ (equal to $\frac{Q\delta^4\pi^4}{45}$); the numerator is fourth degree, the denominator is 8th, so it is integrable on $\mathbb{R}$. This implies that, according to the Lebesgue dominance theorem, we can move the limit from (30) under the integral sign:

$$\sigma_S^2 = \lim_{\xi\to0} I(\xi, \delta) = \int_{-\infty}^{\infty} \lim_{\xi\to0} g(\omega, \xi, \delta) =$$

$$= Q\delta^3\pi^3 \int_{-\infty}^{\infty} \frac{-2z^4\sin^2(z) + 3z^4 - 3\sin^4(z)}{z^8(\cos(2z) + 2)} dz.$$

Now all that's left is to compute the integral:

$$\int_{-\infty}^{\infty} \frac{-2z^4\sin^2(z) + 3z^4 - 3\sin^4(z)}{z^8(\cos(2z) + 2)} dz \triangleq \int_{-\infty}^{\infty} \tilde{g}(z)dz = \dots$$

Again, we use tools from complex analysis. To calculate it, we consider the contour, illustrated by Figure 5: from $-R$ to $-\varepsilon$ on the real axis, along a small semicircle $\gamma_\varepsilon$ around 0, then again along the real axis from $\varepsilon$ to $R$, and finally along a large semicircle $\Gamma_R$.

The function is analytic on the contour, so we can apply the residue theorem:

$$\left(\int_{-R}^{\varepsilon} + \int_{\gamma_\varepsilon} + \int_{\varepsilon}^{R} + \int_{\Gamma_R}\right) \tilde{g}(z)dz = 2\pi i \sum_{\text{poles}} \text{Res}(\tilde{g}, \text{pole}).$$

The integrand has a removable singularity at $z = 0$, so it is bounded in the vicinity of 0. Thus, the integral along $\gamma_\varepsilon$ vanishes as $\varepsilon \to 0$. The integral along $\Gamma_R$ vanishes, since the power of $z$ in the denominator of $\tilde{g}$ is larger than that of the numerator by four (8 vs 4). We are left with:

$$\text{p.v.} \int_{-\infty}^{\infty} \tilde{g}(z)dz = 2\pi i \sum_{\text{poles}} \text{Res}(\tilde{g}, \text{pole}). \tag{32}$$

The integrand has simple poles at $\cos(2z) = -2$:

$$\cos(2z) = -2 \iff z_n^{\pm} \triangleq \frac{\pm i\log(2 - \sqrt{3}) + \pi}{2} + \pi n \ n \in \mathbb{Z}.$$

We are interested in the poles in the upper-half space, so since $\log(2 - \sqrt{3}) < 0$, we choose $z_n \triangleq z_n^-$. Again, using the l'Hopital rule, we can evaluate the residues at these poles; using symbolic calculations to simplify the resulting expressions, we achieve:

$$\text{Res}(\tilde{g}, z_n) = -\frac{9\sqrt{3}i}{8\left(\pi n + \frac{\pi}{2} - \frac{i\log(2-\sqrt{3})}{2}\right)^8}.$$

According to (32), we now need to sum these residues:

$$\sigma_S^2 = Q\delta^4\pi^3 2\pi i \sum_{n=-\infty}^{\infty} \left(-\frac{9\sqrt{3}i}{8\left(\pi n + \frac{\pi}{2} - \frac{i\log(2-\sqrt{3})}{2}\right)^8}\right) \tag{33}$$

The sum from (33) can be calculated analytically, using complex analysis. We introduce the notation $c \triangleq \frac{\pi}{2} - \frac{i \log\left(2-\sqrt{3}\right)}{2}$ for convenience:

$$\sum_n \frac{1}{(\pi n + c)^8} = \dots,$$

The sum appears as the sum of residues of the function:

$$f_{R8}(z) \triangleq \frac{\cot(z)}{(z+c)^8}.$$

Indeed, the residues at poles of $\cot$, for $z = \pi n \; n \in \mathbb{Z}$, are:

$$\text{Res}(f_{R8}, n) = (z - \pi n)\frac{\frac{1}{z-\pi n} + \mathcal{O}(z - \pi n)}{(\pi n + c)^8} = \frac{1}{(\pi n + c)^8}.$$

According to the residue theorem, their sum is minus the residue of the remaining pole at $-\frac{c}{\pi}$:

$$\sum_n \frac{1}{(\pi n + c)^8} = -\text{Res}\left(f_{R8}, -\frac{c}{\pi}\right). \tag{34}$$

It is an 8th-order pole, so its residue is given by:

$$\text{Res}(f_{R8}, 0) = \frac{1}{7!} \lim_{z \to -c/\pi} \frac{d^7}{dz^7}(z^8 f_{R8}(z)) = -\frac{16}{567}.$$

Substituting it first into (34), and then into (33), we get the final result:

$$\sigma_S^2 = Q\delta^4 \pi^3 2\pi \frac{9\sqrt{3}}{8}\frac{16}{567} = \frac{4\sqrt{3}\pi^4 Q\delta^4}{63}.$$

$\square$

We devised a tight error bound for the interval error: indeed, the expression from Lemma 4.6 turns to equality for uniform sequences. It is also important to highlight, that pointwise bounds exist for the non-GP, deterministic setting (and can be plugged into (9), supposing that $\mathbf{x}$ is degenerate) Hall & Meyer (1976); De Boor & De Boor (1978), the optimal one being:

$$\|\hat{\mathbf{x}}(t) - \mathbf{x}(t)\| \leq \frac{5}{384}|\delta|^4\|\mathbf{x}^{(4)}\|.$$

However, they are not tight: the actual error evidently depends on the distance to the closest observation, which is impossible for $\bar{\sigma}_{\text{PW}}$ since it does not depend on time.

### B.5.3 PROOF OF SPECIAL CASE OF ASSUMPTION B.5.

Consider an exponential covariance function $k(x,y) = \exp\left(-|x-y|\right)$. If $y > x$ and we consider $y' = y + s$ for $s > 0$, then $k(x, y') = k(x,y)\exp(-s)$. Below we denote $\delta = \exp(-s)$. By construction $\delta < 1$.

We construct optimal Gaussian process prediction at a point $x$. Our sample of observations consists of two sets of points: $X_1 = \{x_1, \dots, x_n\}$, $X_2 = \{x_{n+1}, \dots, x_m\}$. For our design of experiments, it holds that for $i > n \; x_i > x_k, k = \overline{1,n}, \; x_i > x$. We shift $X_2$ by $s > 0$ to get $X_2^s = \{x_{n+1} + s, \dots, x_m + s\}$.

Let us define the covariance matrices involved in risk evaluation. Self-covariances at $X_1$ is $A$, self-covariances defined at $X_2$ is $D$, and cross-covariances between $X_1$ and $X_2$ is $B$. Then, after changing $X_2$ to $X_2^s$, $A$ and $D$ don't change, and $B$ is multiplied by $\delta$.

Let us denote the vector of covariances between $x$ and $X_1$ and $X_2$ as $\mathbf{a}$ and $\mathbf{d}$ correspondingly. After changing $X_2$ to $X_2^s$, $\mathbf{a}$ doesn't change, and $\mathbf{d}$ is multiplied by $\delta$.

The squared risk for the prediction at point $x$ has the form:

$$\sigma^2(x) = k(x,x) - \mathbf{k}^T K^{-1} \mathbf{k}.$$

Here $\mathbf{k}$ is a concatenation of $\mathbf{a}$ and $\mathbf{d}$, and

$$K = \begin{pmatrix} A & B \\ B^T & D \end{pmatrix}.$$

After shifting of $X_2$, $\mathbf{k}_\delta$ is a concatenation of $\mathbf{a}$ and $\delta\mathbf{d}$, and

$$K_\delta = \begin{pmatrix} A & \delta B \\ \delta B^T & D \end{pmatrix}.$$

and the corresponding risk:

$$\sigma_\delta^2(x) = k(x, x) - \mathbf{k}_\delta^T K_\delta^{-1} \mathbf{k}_\delta.$$

Using the block-inversion formula, we get:

$$K_\delta^{-1} = \begin{pmatrix} A^{-1} + \delta^2 A^{-1} B S^{-1} B^T A^{-1} & -\delta A^{-1} B S^{-1} \\ -\delta S^{-1} B^T A^{-1} & S^{-1} \end{pmatrix},$$

here $S = D - \delta^2 B^T A^{-1} B$.

Then,

$$\mathbf{k}^T K^{-1} \mathbf{k} = \mathbf{a}^T A^{-1} \mathbf{a} + (B^T A^{-1} \mathbf{a} - \mathbf{d})^T S^{-1} (B^T A^{-1} \mathbf{a} - \mathbf{d}),$$

$$\mathbf{k}_\delta^T K_\delta^{-1} \mathbf{k}_\delta = \mathbf{a}^T A^{-1} \mathbf{a} + (B^T A^{-1} \mathbf{a} - \mathbf{d})^T \delta^2 S^{-1} (B^T A^{-1} \mathbf{a} - \mathbf{d}).$$

It is clear that $\delta^2 S^{-1} = \left( \frac{D}{\delta^2} - B^T A^{-1} B \right)^{-1}$.

Let us prove the following Lemma:

**Lemma B.12.** *Consider a positive definite matrix $D$ and $0 < \delta < 1$. Then for an arbitrary vector $\mathbf{x}$ and a symmetric matrix $U$ such that $(D - U)^{-1}$ and $(D/\delta^2 - U)^{-1}$ exist, it holds that:*

$$\mathbf{x}^T (D - U)^{-1} \mathbf{x} \geq \mathbf{x}^T \left( \frac{D}{\delta^2} - U \right)^{-1} \mathbf{x}.$$

*Proof.* For an arbitrary vector $\mathbf{x}$ let us define the function:

$$F(\delta) = \mathbf{x}^T \left( \frac{D}{\delta^2} - U \right)^{-1} \mathbf{x}.$$

Using the derivative of the inverse matrix formula:

$$\frac{\partial F(\delta)}{\partial \delta} = 2\mathbf{x}^T \left( \frac{D}{\delta^2} - U \right)^{-T} \frac{D}{\delta^3} \left( \frac{D}{\delta^2} - U \right)^{-1} \mathbf{x}.$$

Let $\tilde{\mathbf{x}} = \left( \frac{D}{\delta^2} - U \right)^{-1} \mathbf{x}$. Then,

$$\frac{\partial F(\delta)}{\partial \delta} = 2\tilde{\mathbf{x}}^T \frac{D}{\delta^3} \tilde{\mathbf{x}}.$$

Given that $D$ is a positive definite matrix, it holds that:

$$\frac{\partial F(\delta)}{\partial \delta} > 0.$$

So, the proposition of Lemma holds, as with decreasing $\delta$, we decrease the functional value given that the derivative is positive. $\qquad\square$

Substituting $U = B^T A^{-1} B$ and applying Lemma B.12, we get:

$$\mathbf{k}^T K^{-1} \mathbf{k} = \mathbf{a}^T A^{-1} \mathbf{a} + (B^T A^{-1} \mathbf{a} - \mathbf{d})^T S^{-1} (B^T A^{-1} \mathbf{a} - \mathbf{d}) \geq$$

$$\mathbf{k}_\delta^T K_\delta^{-1} \mathbf{k}_\delta = \mathbf{a}^T A^{-1} \mathbf{a} + (B^T A^{-1} \mathbf{a} - \mathbf{d})^T \delta^2 S^{-1} (B^T A^{-1} \mathbf{a} - \mathbf{d}).$$

So, $\sigma_\delta^2(x) > \sigma^2(x)$, meaning that the risk increases after the shift.

### B.5.4 VALIDATION OF ASSUMPTION B.5

The assumption was tested for the common quadratic kernel:

$$K(\delta t) = e^{-(\delta t)^2/2}.$$

The procedure is outlined in Algorithm 1 and is also available as a Python script in our paper's repository. It randomly chooses sequence length, sequence times, displacement amount, and location, and calculates the variance of a Gaussian Process fitted to the generated sequence at a random time. Finally, it asserts that the sequence with increased intervals has an error bigger than the initial one. The loop successfully runs for 1000 iterations, suggesting that the hypothesis typically holds under natural settings.

---

**Algorithm 1** Monte Carlo hypothesis testing.

---

**loop**
    $n \leftarrow \text{randint}(5, 300)$
    $r \leftarrow \text{rand}()$
    **mark** *iteration-start*
    $\{t_k\}_{k=1}^n \leftarrow \text{sort}(\text{rand}(n))$
    $t_1 \leftarrow 0$
    $t_n \leftarrow 1$
    $i \leftarrow \text{randint}(1, N-1)$
    $\{\tilde{t}_k\}_k \leftarrow \{t_1, \ldots, t_{i-1}, t_i + r, \ldots, t_n + r\}$
    $K \leftarrow \text{cov}(\{t_i - t_j\}_{i,j})$
    $\tilde{K} \leftarrow \text{cov}(\{\tilde{t}_i - \tilde{t}_j\}_{i,j})$
    $t \leftarrow \text{rand}()$
    $\tilde{t} \leftarrow t$
    **if** $t_i < t$ **then**
        $\tilde{t} \leftarrow t + r$
    **end if**
    **if** $\text{Singular}(K)$ **or** $\text{Singular}(\tilde{K})$ **then**
        **goto** *iteration-start*
    **end if**
    $k \leftarrow \text{cov}(\{t_k - t\}_k)$
    $\tilde{k} \leftarrow \text{cov}(\{\tilde{t}_k - \tilde{t}\}_k)$
    $D \leftarrow \text{cov}(0) - \text{quad}(K, k)$
    $\tilde{D} \leftarrow \text{cov}(0) - \text{quad}(\tilde{K}, \tilde{k})$
    **assert** $D < \tilde{D}$
**end loop**

---

### B.6 LONG-TERM MEMORY

**Theorem** (4.8). *For the ODE* (2) *under Assumptions 4.1, 4.2, the following holds:*

$$\frac{d}{d\tau} \left\| \frac{d\mathbf{h}(t+\tau)}{dh_i(t)} \right\|_2 < 0.$$

*Proof.* First, we denote $\chi \triangleq \frac{d\mathbf{h}(t+\tau)}{dh_i(t)}$. Instead of considering $\frac{d}{d\tau}\|\chi\|_2$, we consider the derivative of the squared norm $\frac{1}{2}\frac{d}{d\tau}\|\chi\|_2^2$. These two expressions will have the same sign, and the derivative of the squared norm can be conveniently rewritten:

$$\frac{1}{2}\frac{d}{d\tau}\|\chi\|_2^2 = \left( \frac{d}{d\tau}\frac{d\mathbf{h}(t+\tau)}{dh_i(t)}, \chi \right).$$

Since the functions we are working with are continuous, we may re-order the derivative, achieving:

$$\left( \frac{d}{d\tau}\frac{d\mathbf{h}(t+\tau)}{dh_i(t)}, \chi \right) = \left( \frac{d}{dh_i(t)}\frac{d\mathbf{h}(t+\tau)}{d\tau}, \chi \right). \tag{35}$$

Now we can expand $\frac{d\mathbf{h}(t+\tau)}{d\tau}$, using (2), and differentiate it w.r.t. $h_i(t)$:

$$\frac{d}{dh_i(t)}\frac{d\mathbf{h}(t+\tau)}{d\tau} = \frac{d}{dh_i(t)}\left(a\mathbf{f}_\theta(\hat{\mathbf{x}}(t+\tau), \mathbf{h}(t+\tau)) - b\mathbf{h}(t+\tau)\right) = aJ\chi - b\chi,$$

where $J$ is the Jacobain of $\mathbf{f}_\theta$ w.r.t. $\mathbf{h}$. Plugging this into (35), we get:

$$\frac{1}{2}\frac{d}{d\tau}\|\chi\|_2^2 = (aJ\chi - b\chi, \chi) = a\chi^T J^T \chi - b\|\chi\|_2^2. \tag{36}$$

According to Assumption 4.1, the spectral norm of $J$ is no greater than $L_h$. Consequently, the maximum eigenvalue of $J$ is also $\leq L_h$. We know from linear algebra that a quadratic form cannot stretch a vector's norm more than its max eigenvalue, so:

$$\chi^T J^T \chi \leq L_h \|\chi\|_2^2. \tag{37}$$

Finally, putting (36) together with (37), we achieve:

$$a\chi^T J^T \chi - b\|\chi\|_2^2 \leq L_h a\|\chi\|_2^2 - b\|\chi\|_2^2 = \|\chi\|_2^2(L_h a - b).$$

This expression is negative, according to (3). □

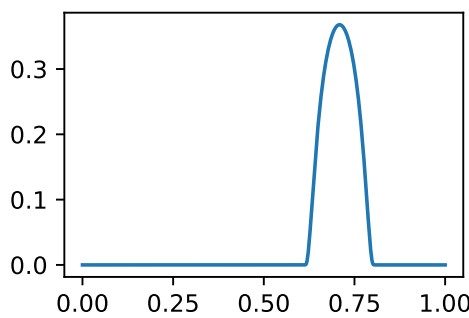

Figure 6: A sample from the bump dataset

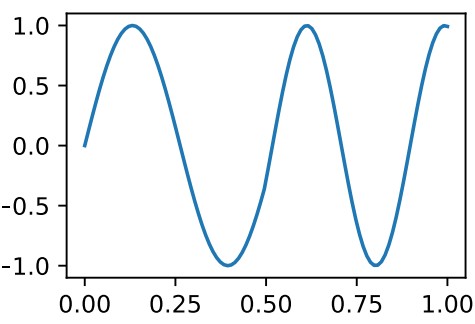

Figure 7: Sine-Mix dataset sample.

## C  EXPERIMENTS

### C.1  DATASETS

Below is a detailed description of each dataset:

- The PhysioNet Sepsis [6] dataset Reyna et al. (2020), released under the CC-BY 4.0 license. Most of the values are missing, with only 10% present. Sequences are relatively short, ranging from 8 to 70 elements, with a median length of 38. The target variable indicates whether the patient developed sepsis, meaning it's **binary** and highly unbalanced, so we measure AUROC. This dataset is also used by the original NCDE paper Kidger et al. (2020), making it a valuable addition to our benchmark. Following the NCDE paper, we only consider the first 72 hours of a patient's stay.

- Two datasets from the Time Series Classification archive [7] Middlehurst et al. (2024) (with no explicit license specified): the U-Wave Gesture Library Liu et al. (2009) and the Insect Sound dataset[8]. These datasets were processed identically: the timestamps were taken to be evenly spaced between 0 and 1, and 30% of each sequence was replaced with NaNs. They both have a **multiclass** label, so we use the Accuracy metric.

- A synthetic Pendulum dataset, with the dampening coefficient as the target Osin et al. (2024) (with no explicit license specified) — **regression** task. Sequences are irregular, containing 10% missing values, with lengths varying from approximately 200 to 400 elements, the median length being 315.

- The Pendulum (regular and irregular versions) and Sine-2 datasets with forecasting as the target, see Appendix C.8.

- The Bump dataset for Section 3. The task is **binary classification**: between bump functions (38) with $\zeta = 20$ and constant-zero functions. Figure 6 demonstrates a positive sample from this dataset.

- The SineMix dataset. The task is to predict the frequency of the first of two sine waves, joined at mid-point as in Figure 7. Consequently, this is a **regression** label, we use $R^2$ score.

$$\psi_\zeta(x) = \begin{cases} \exp\left(\frac{1}{(\zeta x)^2 - 1}\right) & \text{if } |(\zeta x)| < 1; \\ 0 & \text{if } |(\zeta x)| \geq 1 \end{cases} \tag{38}$$

The choice of metrics and activation functions for each dataset is dictated by the nature of the task. The correspondence between tasks, activation functions, and metrics is given by Table 8.

---

[6] https://physionet.org/content/challenge-2019/1.0.0/

[7] https://www.timeseriesclassification.com/

[8] https://www.timeseriesclassification.com/description.php?Dataset=InsectSound

Table 8: Table with Metrics and Activations, corresponding to various tasks.

| Task | Binary Classification | Multiclass Classification | Regression |
|---|---|---|---|
| Activation | Sigmoid | SoftMax | None |
| Metric | AUROC | Accuracy | $R^2$ Score |

This choice of datasets allows us to test the considered models in multiple challenging areas: long sequences, irregular sequences, and sequences with many missing values and scarce data. The code for generating each dataset is provided in our repository.

## C.2 BACKBONES

**RNN.** The recurrent networks are all single-layer and one-directional, with 32 hidden channels. For GRU Chung et al. (2014) we use the standard PyTorch implementation:

$$\mathbf{r} = \sigma(W_{ir}\mathbf{x} + b_{ir} + W_{hr}\mathbf{h} + b_{hr});$$
$$\mathbf{z} = \sigma(W_{iz}\mathbf{x} + b_{iz} + W_{hz}\mathbf{h} + b_{hz});$$
$$\mathbf{n} = \tanh(W_{in}\mathbf{x} + b_{in} + \mathbf{r} \odot (W_{hn}\mathbf{h} + b_{hn}));$$
$$\tilde{\mathbf{h}} = (1 - \mathbf{z}) \odot \mathbf{n} + \mathbf{z} \odot \mathbf{h}.$$

We use the standard GRU notation: $\mathbf{h}$ for the previous hidden state, $\mathbf{x}$ for the current input, $\mathbf{h}'$ for the new hidden state; $\odot$ denotes the Hadamard product. For the vanilla RNN, we use a simple one-layer tanh-activated network:

$$\tilde{\mathbf{h}} = \tanh(W_{ih}\mathbf{x} + b_{ih} + W_{hh}\mathbf{h} + b_{hh}).$$

**Transformers.** We use the Pytorch implementation of the Transformer Encoder layer. The two versions differ in their positional embeddings: RoFormer uses rotary embeddings Su et al. (2024) (provided by TorchTune), and TempFormer uses temporal sine-based embeddings, as proposed by the Transformer Hawkes paper Zuo et al. (2020).

**Mamba.** The Mamba model we use is Mamba2 Gu & Dao (2023) from the Mamba SSM library [9], which we employ without significant modifications.

**Neural CDE, RDE.** To reproduce the results as closely as possible, we use TorchCDE [10] for the Neural CDE method Kidger et al. (2020). Specifically, we implemented the example for irregular sequence classification from their repository. For Neural RDE Morrill et al. (2021b), we pre-process the data using the original Signatory package, provided by the authors [11], with depth set to 2.

**GRU-ODE.** As the model for GRU-ODE De Brouwer et al. (2019), we simply take a Neural CDE with the Sync-NF vector field, without time scaling.

**DeNOTS.** DeNOTS is slower than other non-ODE methods; however, we argue this is an implementation issue. Neural ODEs received much less attention than State-Space Models, Transformers, or RNNs. We use the TorchODE library Lienen & Günnemann (2022). Although it is faster than the original implementation Chen et al. (2018), it still requires work, being built almost single-handedly by its main contributor.

Backpropagation is done via the AutoDiff method, which is faster than Adjoint backpropagation. However, it is also more memory-consuming. We fix the tolerance to $10^{-3}$ and use the adaptive DOPRI5 solver. The normalizing constant $M$ we use for time scaling ($t_k \leftarrow \frac{D}{M}t_k$) is set to the median size of the timeframe across the dataset. Our version of cubic spline interpolation skips NaN values, interpolating only between present ones. All-zero channels are set to a constant zero. For the DeNOTS versions that use the GRU Cell as their dynamics function:

---

[9]https://github.com/state-spaces/mamba
[10]https://github.com/patrick-kidger/torchcde
[11]https://github.com/patrick-kidger/signatory

- For no negative feedback, we use the GRU as-is;
- For Anti-NF, we pass $-\mathbf{h}$ instead of $\mathbf{h}$ to a standard PyTorch GRU.
- For Sync-NF, we subtract the current hidden from the derivative.

GRU turns out to be a surprisingly convenient framework for incorporating negative feedback.

For the non-GRU-based vector fields, specifically Tanh and ReLU, we used two consecutive linear layers separated by the corresponding activations. The tanh model also includes an activation at the end. The ReLU version does not have a final activation; otherwise, it could not decay any hidden-state components.

**Model size.** All the considered models are small, with hidden sizes fixed to 32. This is done deliberately to facilitate reproducibility and reduce experimentation time. The remaining hyperparameters are chosen so that the total number of learnable parameters is comparable to or greater than that of DeNOTS: everywhere except Sepsis that means setting the number of layers to 1. On Sepsis, we use 4 layers for Transformers and 32 layers for Mamba. The specific numbers of parameters for each model on each of our main datasets are provided in Table 9.

Table 9: The number of parameters for each model on all our main datasets.

|             | DeNOTS | GRU    | NCDE    | NRDE  | Mamba  | RoFormer | TempFormer |
|-------------|--------|--------|---------|-------|--------|----------|------------|
| Pendulum    | 3552   | 3552   | 4352    | 7616  | 8120   | 137664   | 137664     |
| UWGL        | 3648   | 3648   | 5440    | 11968 | 8152   | 137504   | 137504     |
| InsectSound | 3456   | 3456   | 3264    | 4352  | 8088   | 137632   | 137632     |
| Sepsis      | 339264 | 339264 | 3809088 | ✗     | 367776 | 662080   | 662080     |

## C.3 PIPELINE

We provide our repository with automated scripts to download and preprocess all datasets, as well as to train and evaluate all the considered models, using popular frameworks such as Pytorch Ansel & team (2024) (Caffe2 license), Pytorch Lightning Falcon & The PyTorch Lightning team (2019) (Apache License 2.0) and Hydra Yadan (2019) (MIT License) to make the process familiar to most AI researchers. The YAML configs, containing all the hyperparameters, are also provided in the repository to facilitate reproducibility. The documentation clearly explains the steps necessary to reproduce all considered experiments.

### C.3.1 TRAINING

All training uses the Adam method Kingma (2014), with the learning rate fixed to $10^{-3}$ and other parameters left default. The whole model, consisting of the head and the backbone, is trained end-to-end. The head consists of a linear layer and an activation function. We do not set an upper bound for epochs, stopping only when the validation metric stops improving. The choice of head's activation, loss, and specific metrics depends on the considered task, as outlined in Table 10.

Table 10: Head activation, loss, and metric choice for each downstream task.

| Task       | Activation | Loss                 | Metric   |
|------------|------------|----------------------|----------|
| Regression | -          | Mean squared error   | $R^2$    |
| Binary     | Sigmoid    | Binary cross-entropy | AUROC    |
| Multiclass | Softmax    | Cross-entropy        | Accuracy |

### C.3.2 EMBEDDINGS

Before passing $\{\mathbf{x}_k\}_k$ to the backbone, we apply Batch Normalization Ioffe & Szegedy (2015). For a fairer baseline comparison, time intervals $t_k - t_{k-1}$ are also included in the embeddings in the same way as other numerical features.

Table 11: Main results on the four considered datasets. Best models are highlighted in **bold**, second-best are underlined. Values are highlighted identically, if their difference is less than half of their joint variance ($\frac{1}{2}\sqrt{\sigma_1^2 + \sigma_2^2}$). We present an average over three runs. For the two SNCDEs (Scaled Neural CDEs), specifically Sync-NF and DeNOTS (Anti-NF), $D$ is selected based on the validation set.

| Backbone | UWGL | InsectSound | Pendulum | Sepsis |
|---|---|---|---|---|
| GRU | $0.5 \pm 0.1$ | $0.3 \pm 0.2$ | $0.73 \pm 0.03$ | $0.838 \pm 0.004$ |
| TempFormer | $0.78 \pm 0.03$ | $\mathbf{0.43 \pm 0.02}$ | $0.59 \pm 0.08$ | $0.89 \pm 0.02$ |
| RoFormer | $0.74 \pm 0.04$ | $0.29 \pm 0.02$ | $0.61 \pm 0.02$ | $0.924 \pm 0.004$ |
| Mamba | $0.71 \pm 0.07$ | $0.41 \pm 0.03$ | $0.61 \pm 0.03$ | $0.829 \pm 0.004$ |
| GRU-ODE | $0.4 \pm 0.3$ | $0.18 \pm 0.03$ | $0.6 \pm 0.05$ | $0.925 \pm 0.003$ |
| Neural CDE | $\mathbf{0.82 \pm 0.03}$ | $0.145 \pm 0.001$ | $0.76 \pm 0.02$ | $0.880 \pm 0.006$ |
| Neural RDE | $0.79 \pm 0.03$ | $0.212 \pm 0.004$ | $\mathbf{0.78 \pm 0.03}$ | ✗[a] |
| Sync-NF SNCDE | $\mathbf{0.811 \pm 0.002}$ | $0.39 \pm 0.09$ | $0.77 \pm 0.01$ | $0.932 \pm 0.003$ |
| DeNOTS (ours)[b] | $\mathbf{0.82 \pm 0.03}$ | $\mathbf{0.44 \pm 0.02}$ | $\mathbf{0.79 \pm 0.02}$ | $\mathbf{0.937 \pm 0.005}$ |

[a]Diverges: Neural RDEs cannot handle large numbers of features, as admitted by its authors.
[b]DeNOTS corresponds to the SNCDE with the Anti-NF vector field.

For the Sepsis dataset, the considered numerical features represent medical variables and are thus presumably more complex than the physical coordinate/acceleration data from the other datasets. To account for this, we inflate each of them to a dimension of 100 using a trainable linear layer before the backbone. Besides, this dataset has static features, which we use as the starting points where applicable. Additionally, since Sepsis is closer to event sequences than time series, we fill the NaN values with zeros prior to passing it to the baselines, indicating "no-event" (this includes the SNCDE models).

On all the other datasets, NaN values for the models that do not support missing values were replaced via forward/backward filling.

## C.4 Full Results

In this section, we present the full results of all our models, which were used to build the ranks from Table 7. The results are given in Table 11.

## C.5 Expressivity discussion

Here, we explain why measuring expressivity with the downstream metric is a valid approach. Theoretically, the total error can be decomposed into three terms Gühring et al. (2020):

1. The approximation error. We are forced to consider only a limited parametric family of estimator functions. This directly measures the expressivity of the chosen parametric model, which is the focus of our work.

2. The estimation error. We can only calculate the empirical risk, on a finite and imperfect training set, instead of the true one. This measures the generalization quality you speak of.

3. The training error. The optimization problems in deep learning are usually complex and non-convex. We can only solve such problems approximately, via iterative techniques with a finite number of steps.

To minimise the influence of the training error, we do not limit the number of epochs, stopping optimization only when the validation metric stops increasing.

**Overfitting.** Next, it is the consensus that the more expressive models suffer more from overfitting Hawkins (2004), i.e. generalize worse. We verify that our models adhere to this by performing the following experiment. We construct a smaller version of the Pendulum dataset; its training set is 1/32 of the original one, and compare the DeNOTS model with various values of $D$ on this benchmark. The results are provided in Table 12. The overfitting effect is observed very clearly:

the "shallow" $D = 1$ model achieves higher metrics on the test set than $D = 10$, and the ranking is reversed on the train set.

Table 12: Results on the Pendulum small dataset for two versions of the DeNOTS model with various time scales ($D = 1, 10$).

| Scale | Test R2 | Test MSE | Train R2 | Train MSE |
|-------|---------|----------|----------|-----------|
| $D = 1$ | **0.479 ± 0.006** | **0.178 ± 0.002** | 0.62 ± 0.096 | 0.124 ± 0.029 |
| $D = 10$ | 0.437 ± 0.049 | 0.192 ± 0.017 | **0.738 ± 0.115** | **0.089 ± 0.043** |

To sum up, we have ruled out the training and estimation errors in our NFE-Metric correlation experiments, so we conclude that high correlation implies that time scaling benefits expressivity.

## C.6 NFE-Correlation experiments

This section focuses on the relationship between the Number of Function Evaluations (NFE) logarithm and the respective metrics. In addition to Pearson correlation from Table 6, measuring linear dependence, we also present Spearman correlation in Table 13, measuring monotonicity. Finally, the log-NFE-metric graphs in Figures 8, 9 illustrate these relationships. The best results of each model are displayed in Table 14.

Table 13: Spearman correlation test between log-NFE and metric, for various vector fields. The first number indicates the statistic for reducing tolerances, the second one — the statistic for increasing time scale. Values $\geq 0.7$ are highlighted in bold.

|  | Pendulum | Sepsis | Sine-2 |
|--|----------|--------|--------|
| Tanh | 0.1/-0.2 | 0.2/**0.7** | 0.6/0.6 |
| ReLU | 0.4/-0.7 | 0.4/0.03 | 0.1/0.5 |
| No NF | 0.4/-0.7 | 0.4/-0.6 | 0.3/-0.3 |
| Sync-NF | 0.4/**0.7** | 0.4/-0.4 | 0.4/**0.9** |
| Anti-NF | 0.5/**0.9** | 0.5/**0.7** | 0.3/**1.0** |

Our model shows excellent performance on all the presented benchmarks. As for the other approaches:

- Increasing tolerance to increase NFE does not reliably improve the models' performance; the NFE-metric correlation is mostly weak.

- Increasing depth for non-stable vector fields does not consistently increase expressiveness.

- The Sync-NF seems to suffer from forgetfulness on larger depths for Sepsis (especially evident from Figure 8).

Notably, the ReLU-T model performs surprisingly well on Sepsis, beating even Anti-NF-S. However, ReLU-T's performance on Pendulum and Sine-2 is inferior, and its NFE-metric correlation is low even on Sepsis, so this does not invalidate our conclusions. We argue that although ReLU activations can lead to dying gradients, potentially destabilizing training, and although the unbounded nature of this vector field (VF) may in principle cause exponential growth in the hidden state, such behavior is not inevitable. The Sepsis dataset is somewhat atypical: approximately *90% of the observations are missing*. As a result, the effective sequence length is substantially shorter, which alleviates some stability concerns, allowing the model to increase $\ell_2$ weight norms without consequence.

## C.7 Negative Feedback Strength

In this section, we analyse whether the equation (3) holds for Anti-NF or Sync-NF in practice. Specifically, we will use the following lemma:

**Lemma C.1.** *If the Jacobian $J \triangleq \left( \frac{\partial \mathbf{f}_i}{\partial \mathbf{h}_j} \right)_{i,j}$ satisfies:*

$$\frac{\|J\mathbf{h}\|}{\|\mathbf{h}\|} \leq L_h, \ \forall \mathbf{h} \in \mathcal{H},$$

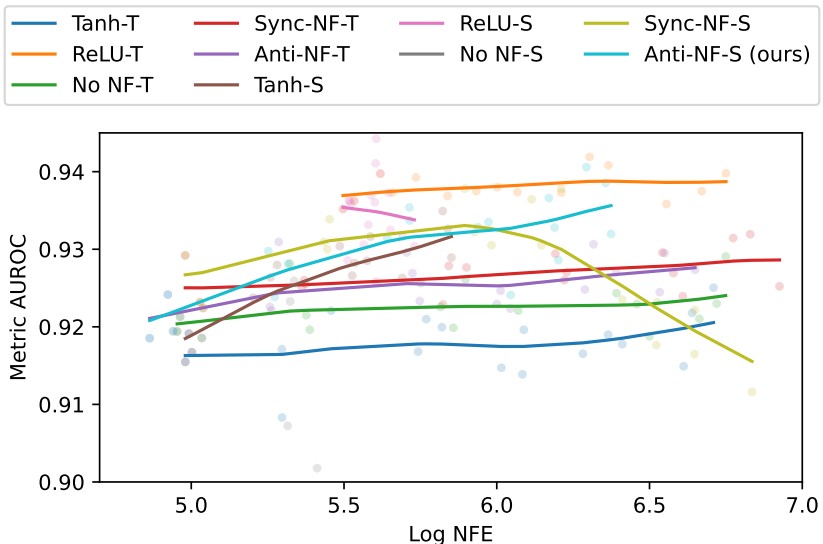

Figure 8: AUROC vs Log-NFEs on the **Sepsis** dataset for various methods of increasing NFEs (-T for lowering tolerance, -S for increasing time scale); for various vector fields (Tanh, ReLU — MLP with Tanh and ReLU activations respectively; No NF — vanilla GRU vector field, Sync NF — GRU-ODE vector field, Anti NF — our version). The curves were drawn via Radial Basis Function interpolation.

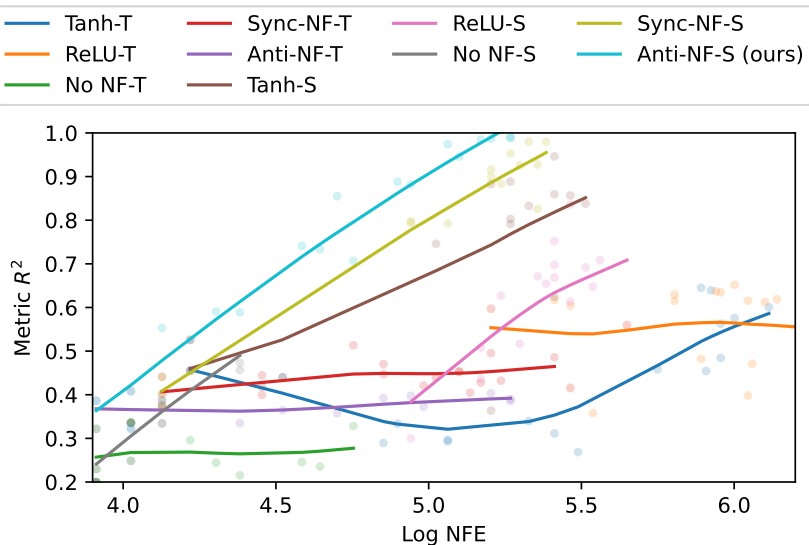

Figure 9: $R^2$ vs Log-NFEs on the **Sine-2** dataset for various methods of increasing NFEs (-T for lowering tolerance, -S for increasing time scale); for various vector fields (Tanh, ReLU — MLP with Tanh and ReLU activations respectively; No NF — vanilla GRU vector field, Sync NF — GRU-ODE vector field, Anti NF — our version). The curves were drawn via Radial Basis Function interpolation.

Table 14: Best results of each vector field in the ablation study. The winner is highlighted in bold, the second-best is underlined; if the distance to (second-)best is less than half the joint variance ($\sqrt{\sigma_1^2 + \sigma_2^2}$), the result is highlighted in the same fashion.

|  | Pendulum | Sepsis | Sine-2 |
| --- | --- | --- | --- |
| Tanh-T | $0.697 \pm 0.007$ | $0.919 \pm 0.002$ | $0.6 \pm 0.04$ |
| ReLU-T | $-20.0 \pm 20.0$ | $\mathbf{0.94 \pm 0.002}$ | $0.58 \pm 0.08$ |
| No NF-T | $0.68 \pm 0$ | $0.923 \pm 0.003$ | $0.27 \pm 0.05$ |
| Sync-NF-T | $0.68 \pm 0.04$ | $0.93 \pm 0.004$ | $0.47 \pm 0.05$ |
| Anti-NF-T | $0.69 \pm 0.03$ | $0.927 \pm 0.003$ | $0.39 \pm 0.02$ |
| Tanh-S | $0.7 \pm 0.09$ | $0.934 \pm 0.002$ | $0.88 \pm 0.06$ |
| ReLU-S | $-50.0 \pm 60.0$ | $0.9364 \pm 0.0007$ | $0.7 \pm 0.1$ |
| No NF-S | $0.65 \pm 0.02$ | $0.92 \pm 0.01$ | $0.46 \pm 0.01$ |
| Sync-NF-S | $0.77 \pm 0.01$ | $0.934 \pm 0.004$ | $0.96 \pm 0.03$ |
| Anti-NF-S | $\mathbf{0.79 \pm 0.03}$ | $\underline{0.937 \pm 0.005}$ | $\mathbf{0.988 \pm 0.002}$ |

*and the set $\mathcal{H}$ is connected, then the function $\mathbf{f}$ is $L_h$-Lipschitz on $\mathcal{H}$.*

It follows from the generalized mean-value theorem (for vector-valued functions with vector arguments).

Consequently, we need to test:

$$\text{For Sync-NF} : \frac{\|J\mathbf{h}\|}{\|\mathbf{h}\|} \leq 1; \tag{39}$$

$$\text{For Anti-NF} : \frac{\|J\mathbf{h}\|}{\|\mathbf{h}\|} a \leq b. \tag{40}$$

For Anti-NF, the expression contains the values $a, b$, which we model with the component-wise mean of $1 - \mathbf{z}, \mathbf{z}$, respectively. We refer to the left-hand sides of (39), (40) as the update strengths, and the right-hand side as the NF strengths. In these terms, we need to test whether the update strength is less than the NF strength.

Figures 10, 11 present our results. The Sync-NF model satisfies our assumptions throughout the trajectory: the update strength is constantly less than 1. The Anti-NF model mostly adheres to (40). However, due to significant variances, the vector field must not always be subject to (40), allowing it to disable the NF effect at will.

## C.8    FORECASTING

Our forecasting task consists of both interpolation and extrapolation, in a simplified setting:

- **Extrapolation.** The second half of the input sequence is discarded;
- **Interpolation.** In the first half, every second element is also discarded.

The task is to reconstruct the discarded elements. Each element is a real number, so forecasting is a regression task in our case. We measure the quality of the solutions using the $R^2$ metric. Specifically, we consider two forecasting datasets: Pendulum-Angles and Sine-2.

### C.8.1    PENDULUM ANGLES

The Pendulum-Angles dataset is the task of forecasting the angle of the pendulum, generated similarly to our synthetic Pendulum dataset. The specifics of the original Pendulum dataset were described in Appendix C.1. To compare our models' ability to handle irregular data, we consider two versions of observation time sampling: with observations on a regular grid or appearing according to a Poisson random process.

The results are presented in Table 15. DeNOTS is best at handling irregular sampling intervals. However, when times are sampled regularly, Neural CDE slightly outperforms our method.

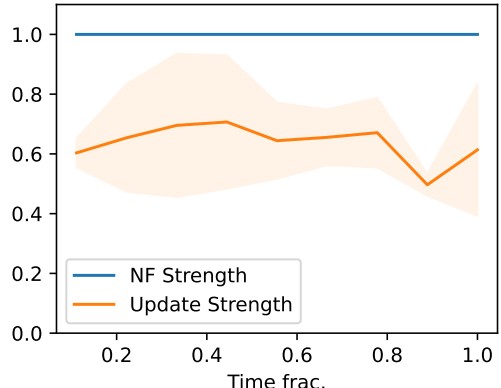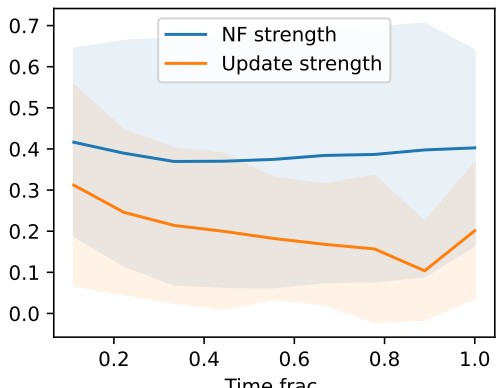

Figure 10: NF and update strength vs fraction of total time passed for the Sync-NF vector field. A constant-1 line represents the NF strength, while for the update strength, we average the results over five runs, illustrating variance via tinting.

Figure 11: NF and update strength vs fraction of total time passed for the Anti-NF vector field. The NF and the update strengths are calculated by averaging over the corresponding values, so we provide variance via tinting for both of them.

Table 15: Results on the Pendulum-Angles dataset.

| Dataset | DeNOTS | Neural CDE | Latent ODE | RoFormer | TempFormer |
|---------|--------|-----------|-----------|----------|-----------|
| Irregular | **0.994 ± 0.001** | 0.985 ± 0.003 | 0.981 ± 0.002 | 0.979 ± 0.001 | 0.961 ± 0.005 |
| Regular | 0.996 ± 0.001 | **0.998 ± 0.001** | 0.996 ± 0.001 | 0.99 ± 0.001 | 0.971 ± 0.003 |

### C.8.2    SINE-2 DATASET

The Sine-2 dataset is a synthetic dataset for forecasting, with each sequence the sum of two sine waves with different frequencies, as illustrated by Figure 12.

This dataset is convenient because it allows us to single out expressivity: we use it in our NFE-Metric correlation experiments. Although it is evidently rather difficult, it is perfectly solvable since it does not contain any noise. Consequently, any increase in expressivity must directly cause an increase in metrics. This is precisely what we observe in Table 6 from the main text, and in Table 13 and Figure 9 from Appendix C.6.

### C.9    FULL ATTACK RESULTS

Here, we provide the complete graphs for both the Change and the Drop attack from Table 5. They are presented by Figures 13, 14. The conclusions are the same as those we provide in the main text, made only more evident by the dynamic. The attacks were each repeated for five different seeds, for five versions of the weights, totalling 25 runs per point, and then averaged over. Using this, we also provide the variance via tinting.

Additionally, we test how the value of $D$ affects the robustness of our models. This is presented in Figures 15, 16. Increasing depth lowers robustness, which is in line with the consensus. Prior work indicates that the more accurate models with higher metrics often perform worse under adversarial attacks than those with lower metrics Su et al. (2018). Increased accuracy often requires increased expressiveness and sensitivity, which directly impacts robustness. Consequently, this demonstration supports our claim that larger values of $D$ correspond to more expressive models.

### C.10    COMPUTATIONAL RESOURCES

Due to the small hidden sizes (32) and limited number of layers (1 in most cases), all the considered models occupy little video memory and fit on an Nvidia GTX 1080 Ti. All the datasets fit into RAM (occupying no more than several Gb on disk). We set a limit of 200Gb RAM in our Docker container,

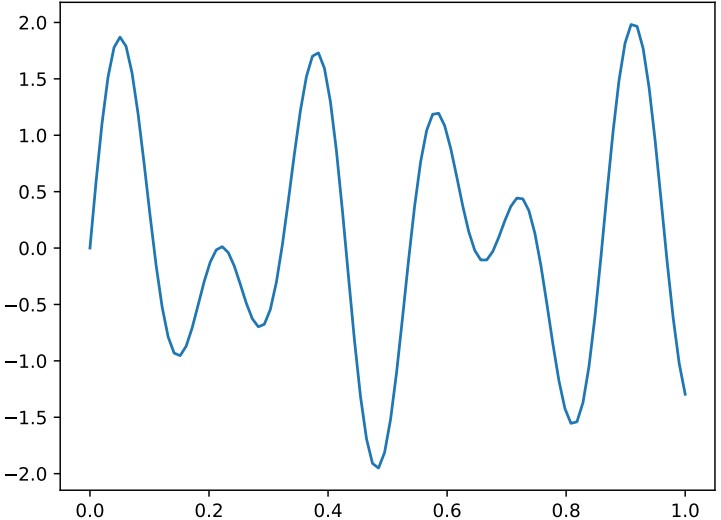

Figure 12: A sample from the Sine-2 dataset.

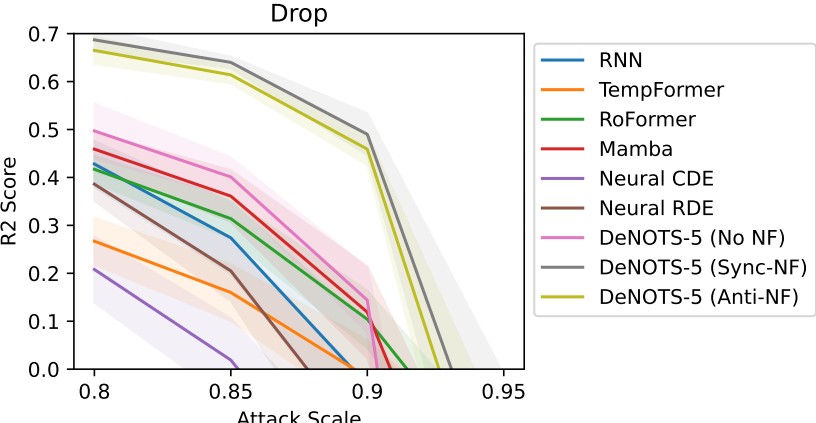

Figure 13: $R^2$ vs fraction of dropped tokens on the Pendulum dataset, on the test set. The dropped tokens are replaced with NaNs. The models are very robust to these attacks, probably because the initial sequences contained NaNs.

however a significantly smaller one would do (we estimate 32Gb should be enough). We also allocate 16 CPUs for our container, but since most of the training happens on a GPU, these are not necessary, one could make do with 4-8 cores.

Each experiment takes from a few minutes to a few hours of compute time, depending on hyperparameters (specifically, $D$ and tolerance for DeNOTS) and the dataset (Pendulum being the most expensive). Overall, we estimate that approximately 100 experiments need to be performed to reproduce our results, with a mean time of 30m, which translates to $\sim 50$ hours of compute on an Nvidia 1080Ti in total. Our cluster houses 3 such video cards, allowing us to perform the computations in parallel, speeding up the process.

Notably, a substantial amount of experiments were not included in the final version due to hypothesis testing and bugs, so the real compute is closer to $\sim 500$ hours.

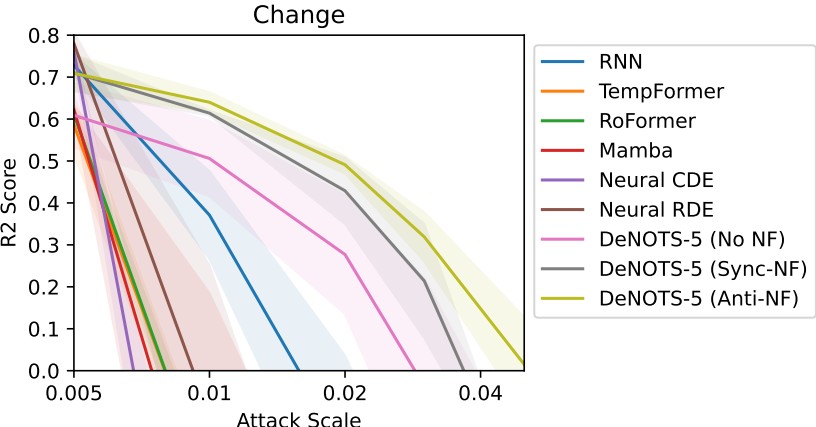

Figure 14: $R^2$ vs fraction of tokens, changed to standard Gaussian noise, on the Pendulum dataset, for the test split. We use the log-scale for the x-axis, because all the considered models are very sensitive to these attacks.

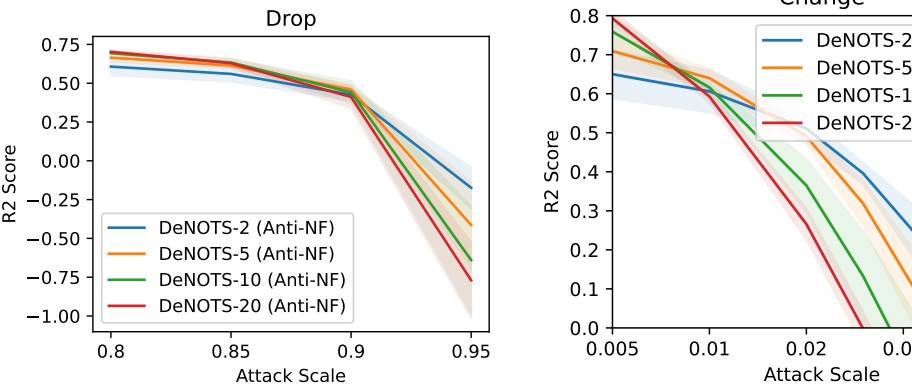

Figure 15: $R^2$ vs fraction of dropped tokens on the Pendulum dataset, on the test set, for various values of $D$. The dropped tokens are replaced with NaNs.

Figure 16: $R^2$ vs fraction of tokens, changed to standard Gaussian noise, on the Pendulum dataset, for the test split, for various values of $D$. We use the log scale for the x-axis because all the considered models are susceptible to these attacks.

