# OpenReview forum: "DeNOTS: Stable Deep Neural ODEs for Time Series"
_ICLR.cc/2026/Conference — ICLR 2026 Poster_

### Official Review · Reviewer_bGMs · 2025-10-17

**Soundness:** 3
**Presentation:** 2
**Contribution:** 3
**Rating:** 6
**Confidence:** 3

**Summary:**

The paper introduces two refinements of Neural CDEs. The first idea is to increase the time horizon of the differential equation in order to increase expressivity. However, this introduces either instability (if the CDE is diverging) or forgetfulness (if the CDE is contractive). The second idea is to add a negative feedback term proportional to the current state, to avoid instability, and to modulate the magnitude of the feedback term through a gate, in order to reduce forgetfulness. This is equivalent to a standard GRU with a minus sign in the gate. Numerical experiments show that the proposed method outperforms both alternative Neural CDE-like models and SSMs. Some theoretical properties are derived, showing improved robustness of the proposed method.

**Strengths:**

Improving the expressivity and stability of neural ODEs/CDEs is an important line of work. The idea proposed by the authors is conceptually simple, yet seems novel and sound. The theoretical and numerical evidence is thorough and convincing. I did not check the proofs.

**Weaknesses:**

The paper discusses many different theoretical results and experiments, which is interesting, but at the same time makes it difficult to follow in some places. For instance:
- the last column of Table 3 is not very clear (I understand that it means that the assumption holds if (and only if?) the condition written is satisfied, but this is not explained). A related comment is that it is stated on line 232 that they analyze empirically whether Assumption 4.2 holds, but on line 243 that is always stands. I did not understand the argument on line 243 and how it relates to the empirical analysis.
- in Theorem 4.5, my understanding is that authors compute how local errors propagate in the ODE, and show a bound on the final state depending on the local errors. I do not understand where the MAP comes into play. It looks to me like $\hat x$ could be any estimator satisfying (8)?
- Section 4.2.1: this section looks interesting but is very difficult to follow. For instance, who is $S$ in Assumption 4.6? What does it mean to assume that the sequence is infinite in our setting? Where does the spectral density of Lemma 4.7 come from?
- Table 5: there is no confidence interval, so it is difficult to assess the robustness of the comparison between Sync-NF and Anti-NF. This makes the hypothesis of lines 405-406 quite brittle.

**Questions:**

See weaknesses.

---

> ### Author Response · Authors · 2025-11-21
>
> 1. On lines 230-231, we write: "we substitute these constraints [from column 4] into [the expression from the assumption] (4) in the last column of Table 3." So, these expressions are made by simple substitutions, i.e. they are equivalent to (4). As for (4) for Sync-NF: on line 243 we explain why this Assumption will likely stand for Sync-NF. In Appendix C6, we demonstrate in practice, that this assumption indeed always stands (what we said on line 232). We will re-write this with increased clarity in light of your concerns.
>
> 2. Theorem 4.5 estimates, how the interpolation error, which comes from the imprefection of our interpolating routine, accumulates in the final hidden state. It can indeed be any estimator, satisfying (8), the MAP condition may be removed from the Theorem; it's there to better illustrate the role of $\hat{\mathbf{x}}$, compared to that of $\mathbf{x}$.
>
> 3. We answer your questions one by one:
>  - $S$ is our input sequence, introduced on line 100 and referred to as "The considered sequence S" in Assumption 4.6.
>  - The infinity assumption is explained in lines 335-338: _"Additionally, prior work (Kolmogorov (1941)) assumes that the considered sequences are infinite. However, the authors of (Zaytsev & Burnaev (2017)) demonstrated that the results for infinite sequences empirically hold for finite ones. Indeed, a typical kernel decays exponentially, so distant observations negligibly impact the resulting error."_
>  - The spectral density from Lemma 4.7 is introduced in the footnote, line 337: _"At the limit $\xi \rightarrow 0$, the mean of the resulting GP tends to a natural cubic spline (Golubev & Krymova (2013)). The parameter $Q$ here represents our a priori beliefs about the data variance."_
>
> 4. The full attack results are provided in Appendix C.8, specifically in Figures 13-16. They illustrate the quality for varying intensity of the attack, with standard deviations illustrated as tinted background around the line.

---

> > ### Comment · Reviewer_bGMs · 2025-11-26
> >
> > I thank the authors for the rebuttal, which answered my questions. I still believe that Section 4.2.1 would benefit from more detailed explanations in order to be understandable by an audience that goes beyond GP specialists. I increased my score.

---

### Official Review · Reviewer_PDGR · 2025-10-30

**Soundness:** 3
**Presentation:** 2
**Contribution:** 2
**Rating:** 4
**Confidence:** 4

**Summary:**

This paper focus on solving downstream tasks for time series with global, sequence-wise targets (binary/multiclass/regression).
The core idea is to deliberately ​scale the integration time horizon​ instead of lowering solver tolerances to increase the model's "depth" (Number of Function Evaluations - NFE), which leads to superior expressiveness.
They also introduced a negative feedback mechanism (derived from control theory) to enhance stability.

**Strengths:**

This article employs a variety of theoretical tools and proposes a straightforward and easy-to-use methodological innovation (time scaling and negative feedback).

**Weaknesses:**

Although this article is based on theoretical analysis, I am somewhat skeptical about whether they can correctly validate its arguments. In my view, it is not entirely logically rigorous. The specific details are listed in the Questions part.

**Questions:**

Theorem 3.1 concludes that scaling T makes the model 'exponentially' more expressive, so it's worth  scaling T.
However, we can also easily reach the following conclusion: scaling T would also 'exponentially' impairs the model's performance. For example, as a well known result [1], for an ODE system, the error of the Euler numerical solution (which is required both in training and inference stages) is $|\varepsilon_n| \leq e^{T L} |\varepsilon_0| + \frac{R}{hL} (e^{TL} - 1)$, which is also exponential of T. In fact, the difficulty of training Neural ODEs or RNNs lies in the need for backpropagation-through-time to track the gradients of weights (e.g., in [2], "Flow-based models were previously limited by inefficient simulation-based training objectives that require an expensive integration of the ODE at training time.").
What I'm concerned about is whether extending the time T will touch upon the core pain points of such models.
So at present, I cannot see that scaling time is an essential improvement. It can certainly enhance the model's fitting ability, but it will also increase the difficulty and error in training and inference for the model (also exponentially).

As for the issue of increasing NFE (network depth) that you mentioned, with the same error tolerance (under the adaptive step-size framework), the size of network weights, and the degree of function stiffness can also affect NFE. If you require a longer integration time to achieve a smoother vector field function, NFE may not necessarily increase, so there may not be a necessary connection between the two. And larger l_2 but small integration times, under the same error tolerance,  I cannot believe there would be any fundamental difference in the results.

[1] Hairer, Ernst, Gerhard Wanner, and Syvert P. Nørsett. Solving ordinary differential equations I: Nonstiff problems. Berlin, Heidelberg: Springer Berlin Heidelberg, 1993.

[2] Tong, Alexander, Nikolay Malkin, Kilian Fatras, Lazar Atanackovic, Yanlei Zhang, Guillaume Huguet, Guy Wolf, and Yoshua Bengio. "Simulation-free schr\" odinger bridges via score and flow matching." arXiv preprint arXiv:2307.03672 (2023).

---

> ### Author Response · Authors · 2025-11-21
>
> ## Rebuttal
>
> 1. Accounting for integration error is indeed important, and adding such theoretical analysis would singificantly strengthen our contribution. We present it below. In short: it turns out, that adding Negative Feedback prevents the numerical integration error from accumulating in the final hidden state.
>
> In essence, Theorem 4.5 comes very close to the point you that raised. It assumes that the derivative is calculated with some error, which stems from approximate interpolation of the input trajectory. If we instead say that this error comes from discretisation during the integration process, we will get the numerical integration error.
>
> We start with two ODEs. The first one is "true", which we cannot fully observe:
>
> $$
> \frac{d\mathbf{h}}{dt} = a \mathbf{f}(\mathbf{x}, \mathbf{h}) - b \mathbf{h}.
> $$
>
> And the second one is its numerical approximation, which we are capable of calculating, often referred to as the "modified vector field" [1]:
>
> $$
> \frac{d\hat{\mathbf{h}}}{dt} = a \mathbf{f}(\mathbf{x}, \hat{\mathbf{h}}) - b \hat{\mathbf{h}} + \delta(t).
> $$
>
> Essentially, the numerical approximation is defined by how our solver models the trajectory. $\delta(t)$ is some error term, introduced at each timestep. It depends on the parameters of the solver. For simplicity, let's assume that $\|\delta(t)\| \le \varepsilon$, for some $\varepsilon > 0$.
>
> Using Assumption 4.2, we may formulate the following fact:
>
> **Numerical Error Theorem:** Under the strong-NF assumption (4.2 in the paper), the distance between the true vector field and the modified one is bounded by a constant, independent of $T$:
> $$
> \| \hat{\mathbf{h}} - \mathbf{h} \| \le \frac{a \varepsilon}{b - a L_h}.
> $$
>
> The proof is provided in the comments below. As you can see, it implies that our NF mechanism is able to stabilise integration error, and to prevent it from growing with increasing time scale.
> It is worth noting, however, that we have indeed observed state explosion for non-NF vector fields, as illustrated in Figure 4.
>
> 2. Now for the connection between NFE and $D$.
> To address your concerns, we provide the table with the Spearman correlations between NFE and D on the Pendulum dataset:
> | Anti-NF | Sync-NF | ReLU | Tanh | No NF |
> |---------|---------|------|------|-------|
> | 0.93    | 0.96    | 0.87 | 0.78 | 0.94  |
>
> As you can see, we observe strong NFE-$D$ correlation.
> While increasing $L_2$ norms for small values of $D$ may be viable, in practice the methods fail to do so due to training instabilities: specifically, Tanh saturation and ReLU dying for large gradients.
>
> Finally, we would like to highlight that, for the reasons outlined above, there is a fundamental difference in the results between small and large timeframes for the same error tolerance.
> This difference is extensively studied in Figures 1, 8, 9 in terms of NFE.
> We additionally provide a Figure to illustrate specifically the $D$-Metric relationship for our model, where we claim that the results improve with increasing timescale:
> https://postimg.cc/NyT5tKrc
>
> ### References:
> [1]: Reich, Sebastian. "Backward error analysis for numerical integrators." SIAM Journal on Numerical Analysis 36.5 (1999): 1549-1570.

---

> > ### Comment · Reviewer_PDGR · 2025-11-23
> >
> > I'm not discussing the second part of your argument, which is about the significance of introducing negative feedback. (Of course introducing negative feedback can indeed suppress errors, limiting the model's expressive power.) What I'm addressing is the first part of your argument—my point is that your theorem doesn't prove the advantage of scaling T, because it also increases training difficulty.
> >
> > Moreover, what I'm questioning is not the relationship between NFE and Depth (isn't Depth characterized by NFE? If not, how do you characterize the Depth of NODE here?). What I'm questioning is that increasing time and reduce $l_2$ does not necessarily increase NFE.

---

> ### Author Response · Authors · 2025-11-21
>
> **Proof of Numerical Error Theorem:**
>
> First, we introduce the following notation, for brevity:
>
> $$
> \Delta_h(t) \triangleq \hat{\mathbf{h}}(t) - \mathbf{h}(t),
> $$
>
> $$
> \Delta_f(t) \triangleq \mathbf{f}(\mathbf{x}(t), \hat{\mathbf{h}}) - \mathbf{f}(\mathbf{x}(t), \mathbf{h}).
> $$
>
> Now, we consider the time-derivative of the squared error norm, multiplied by $\frac12$ for convenience:
>
> $$
> \frac12 \frac{d}{dt}\|\Delta_h\|_2^2 =
> \left( \frac{d\Delta_h}{dt}, \Delta_h \right)
> $$
>
> The derivative term can be simplified, using the ODE's definition:
>
> $$
> \frac{d\Delta_h}{dt} = a\Delta_f - b\Delta_h + a \delta.
> $$
>
> Substituting it, we get:
>
> $$
> \frac12 \frac{d}{dt}\|\Delta_h\|_2^2
> = \left( a\Delta_f - b\Delta_h + a \delta, \Delta_h \right) =
> $$
>
> $$
> = a(\Delta_f, \Delta_h) - b\| \Delta_h \|_2^2 + a (\delta, \Delta_h) \le
> $$
> Now we apply the Cauchy-Schwartz and Lipschitz inequalities:
> $$
> \le a L_h \|\Delta_h \|_2^2 - b \|\Delta_h\|_2^2 + a \delta \| \Delta_h \|_2.
> $$
>
> Putting it together, bounding $\delta$ with $\varepsilon$, we get the following differential inequality:
> $$
> \frac12 \frac{d}{dt} \| \Delta_h \|_2^2 \le \| \Delta_h \|_2^2 (a L_h - b) + a \varepsilon \| \Delta_h \|_2 \triangleq Q(\| \Delta_h \|).
> $$
>
> On the right is a quadratic function of $\| \Delta_h \|_2$ that we denote $Q(\| \Delta_h \|)$.
> Since we assume that $a L_h - b < 0$, it will be negative for large enough $\| \Delta_h \|_2$. Specifically:
>
> $$
> \text{IF}\; \| \Delta_h \| \ge \frac{a \varepsilon}{b - a L_h}\; \text{THEN}\; Q(\| \Delta_h \|) < 0.
> $$
>
> Putting this together with the differential inequality, we get that:
>
> $$
> \text{IF}\; \| \Delta_h \| \ge \frac{a \varepsilon}{b - a L_h}\; \text{THEN}\; \frac12 \frac{d}{dt} \| \Delta_h \|_2^2 < 0.
> $$
>
> Consequently, $\| \Delta_h \|$ cannot become larger than $\frac{a \varepsilon}{b - a L_h}$ as it will immediately start decreasing. This fact proves the proposed bound.

---

> ### Author Response · Authors · 2025-11-23
>
> - Scaling $T$ does not seem to increase training difficulty in practice: we observe that scaling time increases the resulting metrics, while the training routine is kept identical across different values of $D$ (the time scaling factor). The reference from Tong et al. from your initial review only states that the procedure is more computationally intensive, not that these models are more difficult to train. Kindly, elaborate on this point: which training difficulties do you mean in this context?
>  - The numerical error may indeed be a problem. However, we have proved that our method does not accumulate numerical error with NF. If done correctly (Anti-NF), it also does not seem to limit the model's expressiveness, according to our experiments (see Figures 1, 8, 9).
>
> As for your second paragraph: sorry, there seems to be a misunderstanding. By Depth with a capital D, we meant our hyperparameter $D$, responsible for scaling time. We have corrected our comment to reflect that more clearly. Our argument is that time scaling (i.e. increasing $D$) is directly connected to increasing NFE in practice.

---

### Official Review · Reviewer_NLhn · 2025-10-31

**Soundness:** 1
**Presentation:** 2
**Contribution:** 1
**Rating:** 2
**Confidence:** 3

**Summary:**

This paper introduces DeNOTS, a new variant of Neural CDEs for time series. The authors proposes two main modificaitons: time scaling increaisng time interal imprvoe the expressive, while causing exploring vector fields, hence, propose use negative feedback to stablize long horizon integration. They claim that DeNOTS improves expressivlity and stabling as increasing time horizon.

**Strengths:**

try to study an important problem with theoreical motivaiton, to improve expreisivty and stalbity.

**Weaknesses:**

1. I am not convinced by the theoretical results and analysis. Theorem 3.1 shows that a larger integration horizon $T$ leads to a larger $L_F$, but this merely indicates greater output variance, not increased expressivity. To rigorously support the claim that longer integration time enhances expressivity, the authors should establish a functional inclusion relation between model families; e.g., for two horizons $T<T'$, show that the corresponding function classes $H_T \leq H_{T'}$. Without such reasoning, the argument that "increasing $T$ improve expressivity" is not theoretically substantiated.
2. Negative feedback may suppress dynamics rather than enhance expressivity. With the proposed negative-feedback mechanism, increasing $T$ does not always enrich the representation. Consider the scalar linearized dynamics $h' = af(x,h)-b h$ with $f(x,h)\approx \lambda h$. This yields the closed form solution: $h_t = h_0 e^{(a\lambda -b)t}$. With $a+b=1$, if $b$ is small and close to zero, $h_t \propto e^{\lambda t}$ diverges or vanishes exponentially depending on the sign of $\lambda$; if $b$ is large and close to one, $h_t\propto e^{-t}$ decays exponentially to zero; and if $b$ is balanced such that $(a\lambda -b)t\approx 0$, then $h_t$ remain nearly close to its initial $h_0$ with minimal dynamics. In all these regimes, longer $T$ does not meaningfully improve expressivity, contradicting the paper's central claim.
3. The reported gains (1–3 % in R^2 or AUROC) are within the expected variance and are not statistically significant. Given the small benchmark size and lack of confidence intervals, the empirical evidence does not convincingly demonstrate a substantive improvement over existing Neural ODE or CDE baselines.

**Questions:**

See the weakness.

---

> ### Author Response · Authors · 2025-11-14
>
> 1. Indeed, the current version of the paper only establishes that both $H_T$ and $H_{T'}$ are subsets of a Lipschitz class of functions, without a direct subset relation between them. Directly proving that increasing the time-scale broadens this class would significantly strengthen our paper. We provide just that below, along with the corresponding proofs.
>
> First, we revisit and correct some definitions:
>
>  - We change our distance between trajectories to $L_2$, defined as:
> $$
> \rho(\mathbf{x}_1, \mathbf{x}_2; t) = \sqrt{\int_0^t \| \mathbf{x}_1(t) - \mathbf{x}_2(t) \|_2^2}
> $$
>  - We introduce the class of differentially definable mappings from trajectories $\mathcal{F}$. We say that $F_ \mathbf{g} \in \mathcal{F}$, if there exists an ODE with the vector field $\mathbf{g}(\mathbf{x}(t), \mathbf{h}(t)): \mathbb{R}^u \times \mathbb{R}^v \rightarrow \mathbb{R}^v$, with the starting condition $\mathbf{h}(0) = \mathbf{h}_ 0$ and the solution $\mathbf{h}(t): \mathbb{R} \rightarrow \mathbb{R}^v$ :
>  $$
>  \frac{d}{dt} \mathbf{h}(t) = \mathbf{g}(\mathbf{h}(t), \mathbf{x}(t)),
>  $$
>  such that $F_ \mathbf{g}(\mathbf{x}(\cdot), \mathbf{h}_ 0; t) = \mathbf{h}(t)$. Not all functions on trajectories can be represented this way.
>  - We also introduce the class of start-corrected differentially-definable mappings:
>
> $$
> \mathring{\mathcal{F}} = \{\mathring{F}_ \mathbf{g}(\mathbf{x}(\cdot), \mathbf{h}_ 0; t) = F_\mathbf{g}(\mathbf{x}(\cdot), \mathbf{h}_ 0; t) - \mathbf{h}_ 0 | F_\mathbf{g} \in \mathcal{F} \}.
> $$
>
>  Since we set $\mathbf{h}_0=0$ in most practical cases, the distinction is rather technical. This is mostly to account for Lipschitzness w.r.t. $\mathbf{h}$, as we will see below.
>  - Finally, we introduce two ways to constrain this class with Lipschitz constants: with $\mathbf{g}$ being $M_x, M_h$-Lipschitz
>
> $$
> \mathring{\mathcal{F}}_ g(M_ x, M_ h) = \{\mathring{F}_ \mathbf{g} \in \mathring{\mathcal{F}} | \mathbf{g}-M_ x,M_ h \text{-lipschitz w.r.t. } \mathbf{x}, \mathbf{h}\},
> $$
>
>  and with $\mathring{F}_ \mathbf{g}(\ldots; t)$ being $L_ x(t), L_ h(t)$-lipschitz, where $L_ x(t), L_ h(t):\mathbb{R} \rightarrow \mathbb{R}_ +$:
>  $$
>     \mathring{\mathcal{F}}_ F(L_x(\cdot), L_h(\cdot)) = \{\mathring{F}_ \mathbf{g} \in \mathring{\mathcal{F}} | \mathring{F}_ \mathbf{g}-L_x(t), L_h(t)\text{-lipschitz w.r.t. } \mathbf{x}, \mathbf{h}_ 0\}.
>  $$
>  Note, that $\mathring{\mathcal{F}}_g(M_x, M_h)$ represents all mappings that our ODE can learn, given a Lipschitz-constrained Neural Network $\mathbf{g}$, while $\mathring{\mathcal{F}}_F(L_x(\cdot), L_h(\cdot))$ represents all the possible Lipschitz-mappings that we may want to learn.
>
>  Equipped with the above definitions, we can formulate the following theorem, which is a more general version of Theorem 3.1 from the paper:
>
>  **Theorem 3.1 (new):** The classes $\mathring{\mathcal{F}}_g(M_x, M_h)$ and $\mathring{\mathcal{F}}_F(L_x(\cdot), L_h(\cdot))$ are equal, given the following relations between their arguments:
>  $$
>  L_x(t) = M_x \sqrt{\frac{1}{2 M_h}\left(e^{2M_h t} - 1\right)}; L_h(t) = e^{M_h t} - 1.
>  $$
>
> As a corollary, given an $M_x,M_h$-Lipschitz NN $\mathbf{g}$, we can represent all the $L_x,L_h$-Lipschitz (differentially-definable, start-corrected) $F$, and only them. Both $L_x(t)$ and $L_h(t)$ grow monotonically with $t$, with order $\sim e^{M_h t}$. Consequently, increasing $t$ will strictly widen our class of functions (because all $A$-Lipschitz functions are also $B$-Lipschitz for any $A < B$, by definition). So in your notation, we have proved that $H_T \subset H_{T'}$ for all $T < T'$, by tightly establishing $H_T$.

---

> ### Author Response · Authors · 2025-11-14
>
> 2. Yes, the Negative Feedback, especially if it is too strict, as is the case with Sync-NF, will constrain the function class. However, without Negative feedback, errors will accumulate exponentially during integration, so the gains from time scaling will be outweighed by exponential error growth. Anti-NF provides a clean way to a) stabilize integration, but b) does not introduce any significant constraints on the resulting approximator. Anti-NF does not necessarily follow Assumption 4.2, although, in practice, all the beneficial properties of that assumption still stand: specifically, it's robust to attacks and stable, as we demonstrate extensively in our experiments. Specific experiments on Assumption 4.2 may be found in Appendix C.6.
>
> We argue that larger timeframes allow our models to distinguish subtle differences in the input $\mathbf{x}$ better (formally described in terms of Lipschitz constants). The example you provide is not indicative of such expressivity, since it does not depend on the input $\mathbf{x}$ (other than an auxillary dependance on time: both the solution $\mathbf{h}$ and the input $\mathbf{x}$ may depend on $T$). However, it does show that Anti-NF may approximate virtually any function, which aligns with our claims.
>
> 3. Our DeNOTS model is the only one, that shows good stable performance accross all benchmarks. While the baselines indeed catch up in some cases, there is no single baseline that shows good quality all accross our benchmark; each of them fails on one or more datasets. To support this argument, we also provide a **Table with performance ranks (lower is better)**, built on our results:
>
> | Backbone             | UWGL | InsectSound | Pendulum | Sepsis | Mean |
> |----------------------|------|-------------|----------|--------|------|
> | DeNOTS (ours)        | 1    | 1           | 1        | 1      | 1.0  |
> | Sync-NF SNCDE (ours) | 1    | 1           | 2        | 1      | 1.25 |
> | TempFormer           | 4    | 1           | 6        | 5      | 4.0  |
> | Neural CDE           | 1    | 8           | 2        | 5      | 4.0  |
> | Neural RDE           | 1    | 6           | 1        | 9      | 4.25 |
> | RoFormer             | 5    | 5           | 6        | 3      | 4.75 |
> | GRU                  | 8    | 1           | 4        | 7      | 5.0  |
> | Mamba                | 6    | 1           | 6        | 8      | 5.25 |
> | GRU-ODE              | 8    | 7           | 6        | 3      | 6.0  |
> The rank of each backbone is one plus the number of other backbones, the score of which is better according to a one-sided T-test on the corresponding mean and deviations, with the p-value cutoff set to 0.1. Therefore, **lower is better**, a rank of 1.0 means that there are no better baselines, and the one in question is the best.
> Note that Sync-NF SNCDE cannot be considered a baseline, it's modified according to our time-scaling contribution (SNCDE). The corresponding vanilla baseline, GRU-ODE, is at the bottom of the ranking.
>
> Besides, we also provide extensive attack-robustness experiments, which also favour our methods over the baselines.

---

> ### Author Response · Authors · 2025-11-14
>
> **Proof of Theorem 3.1 (new). Part 1**:
>
> 1. First, we prove the relation $\mathring{\mathcal{F}}_g(M_x, M_h) \subseteq \mathring{\mathcal{F}}_F(L_x(\cdot), L_h(\cdot))$. Let's consider an $M_x,M_h$-Lipschitz $\mathbf{g}$, and prove that the corresponding $\mathring{F}$ is $L_x,L_h$-Lipschitz.
>
> Say, we have two trajectories, $\mathbf{x}_1(t),\mathbf{x}_2(t)$, each producing their repsective $\mathbf{h} _{1,2}(t)$, with $\mathbf{h} _{1}(0) \ne \mathbf{h} _2(0)$. The distance between the corresponding $\mathring{F}$-s will be given by:
> $$
>     \| (F(\mathbf{x}_1, \mathbf{h}_0, t) - F(\mathbf{x}_1, \mathbf{h}_0, 0)) - (F(\mathbf{x}_2, \mathbf{h}_0, t) - F(\mathbf{x}_2, \mathbf{h}_0, 0)) \| \triangleq \| \Delta(t) \|,
> $$
> where we denote the expression under the norm by $\Delta$ for brevity.
> The absolute derivative of the norm is always less or equal to the norm of the derivative, which can be proven using the Cauchy-Schwartz inequality:
> $$
> \frac{d}{dt}
> \| \Delta \|_2
> \le
> \left|
> \frac{d}{dt}
> \| \Delta \|_2
> \right|
> =
> \left|
> \frac{d}{dt} \sqrt{\Delta \cdot \Delta}
> \right|
> =
> \frac{ \Delta \cdot  \Delta' }{\| \Delta \|_2}
> \le
> \frac{\| \Delta \|_2 \cdot \| \Delta' \|_2}{\| \Delta \|_2}
> = \| \Delta' \|_2 =
> $$
> Here $\Delta'$ is the derivative of $\Delta$ w.r.t. $t$. We can write it using $\mathbf{g}$, the starting points cancel out:
> $$
> = \|\Delta'(t)\|_2 = \|\mathbf{g}(\mathbf{x}_1(t), \mathbf{h}_1(t)) - \mathbf{g}(\mathbf{x}_2(t), \mathbf{h}_2(t))\|_2 =
> $$
> We add and subtract a "mixed`` term $\mathbf{g}(\mathbf{x}_1(t), \mathbf{h}_2(t))$:
> $$
> = \|\mathbf{g}(\mathbf{x}_1(t), \mathbf{h}_1(t)) - \mathbf{g}(\mathbf{x}_1(t), \mathbf{h}_2(t)) + \mathbf{g}(\mathbf{x}_1(t), \mathbf{h}_2(t)) - \mathbf{g}(\mathbf{x}_2(t), \mathbf{h}_2(t))\|_2 \le
> $$
> The norm of that sum may be bounded using the triangle rule:
> $$
> \le
> \|\mathbf{g}(\mathbf{x}_1(t), \mathbf{h}_1(t)) - \mathbf{g}(\mathbf{x}_1(t), \mathbf{h}_2(t)) \|_2
> +
> \|\mathbf{g}(\mathbf{x}_1(t), \mathbf{h}_2(t)) - \mathbf{g}(\mathbf{x}_2(t), \mathbf{h}_2(t))\|_2 \le
> $$
> And both of these two norm-terms can now be bounded by the Lipschitz property:
> $$
> \le M_x \|\mathbf{x}_1 - \mathbf{x}_2\|_2 + M_h \|\mathbf{h}_1 - \mathbf{h}_2\|_2 =
> $$
> We add and subtract $\mathbf{h}_1(0) - \mathbf{h}_2(0) \triangleq \Delta_0$ in the second term:
> $$
> M_x \|\mathbf{x}_1 - \mathbf{x}_2\|_2 + M_h \|\mathbf{h}_1(t) - \mathbf{h}_2(t) - (\mathbf{h}_1(0) - \mathbf{h}_2(0)) +
> (\mathbf{h}_1(0) - \mathbf{h}_2(0))\|_2 \le
> $$
> We now notice that we can substitute $\Delta$ in the second term, and upper-bound it with the triangle inequality
> $$
> \le M_x \|\mathbf{x}_1 - \mathbf{x}_2\|_2 + M_h \|\Delta\|_2 + M_h \|
> \Delta_0 \|_2,
> $$
> where we denote $\Delta_0 \triangleq \mathbf{h}_1(0) - \mathbf{h}_2(0)$.
>
> Putting the above reasoning together, we are left with the differential inequality:
> $$
> \frac{d}{dt} \| \Delta \| \le M_x \|\mathbf{x}_1 - \mathbf{x}_2\| + M_h \|\Delta\| + M_h \|\Delta_0\|.
> $$
> Using the standard ODE trick, we assume that $\Delta(t) = C(t) e^{M_h t}$, and substitute this into the inequality:
> $$
> \frac{d}{dt} \| \Delta \| = \frac{dC}{dt} e^{M_h t} + C(t) M_h e^{M_h t} \le M_x \|\mathbf{x}_1 - \mathbf{x}_2\| + M_h C(t) e^{M_h t} + M_h \|\Delta_0\|.
> $$
> The terms $M_h C(t) e^{M_h t}$ cancel out:
>
> $$
> \frac{dC}{dt} e^{M_h t} \le M_x \|\mathbf{x}_1 - \mathbf{x}_2\| + M_h \|\Delta_0\|.
> $$
>
> Now, we separate the variables, and integrate from $\tau=0$ to $t$ :
>
> $$
> \int_{\tau=0}^t dC = C(t) - C(0) \le \left( M_x\int_{\tau=0}^t \| \mathbf{x}_1(\tau) - \mathbf{x}_2(\tau) \| e^{-M_h \tau} d\tau \right) +
> $$
>
> $$
> \ + M_h \| \Delta_0 \| \int_{\tau=0}^t e^{-M_h \tau} d\tau.
> $$
>
> The integral without $\mathbf{x}_{1,2}(t)$ can be integrated analytically:
>
> $$
> \int_{\tau=0}^t e^{-M_h \tau} d\tau = \frac{1}{M_h} \left( 1 - e^{-M_h t} \right).
> $$
>
> The integral with $\mathbf{x}_{1,2}$ can be bounded using Cauchy-Schwartz:
>
> $$
> \int_{\tau=0}^t \|\mathbf{x}_1(\tau) - \mathbf{x}_2(\tau)\| e^{-M_h \tau} d\tau \le
> $$
>
> $$
> \le \sqrt{ \int_{\tau=0}^t \|\mathbf{x}_ 1(\tau) - \mathbf{x}_ 2(\tau)\|^2 d\tau } \sqrt{\int_{\tau=0}^t e^{-2M_h \tau} d\tau}
> $$
>
> The first term is the $L_2$-distance we introduced earlier, $\rho$, the second one can be integrated analytically:
>
> $$
> \int_{\tau=0}^t e^{-2M_h \tau} d\tau = \frac{1}{2 M_h} \left( 1 - e^{-2 M_h t} \right).
> $$
>
> As a starting condition, we know that $\Delta(0) = 0$ (since it is start-corrected), so $C(0) = 0$. Now we have an inequality for $C(t)$:
>
> $$
> C(t) \le \| \Delta_0 \|_2 \left( 1 - e^{-M_h t}\right) + M_x \rho (\mathbf{x}_1, \mathbf{x}_2) \sqrt{ \frac{1}{2 M_h} \left( 1 - e^{-2 M_h t} \right)}.
> $$
>
> Multiplying it by $e^{M_h t}$ gives us the inequality for $\Delta(t)$:
> $$
> \|\Delta(t)\| \le \| \Delta_0 \|_2 \left(e^{M_h t} - 1\right) + M_x \rho (\mathbf{x}_1, \mathbf{x}_2) \sqrt{\frac{1}{2 M_h} \left( e^{2 M_h t} - 1 \right)}.
> $$
>
> To get the individual expression for $L_x$, we simply set $\Delta_0$ to zero. To get the expression for $L_h$, we set $\rho$ to zero.

---

> ### Author Response · Authors · 2025-11-14
>
> **Proof of Theorem 3.1 (new). Part 2**:
>
> Now we prove the converse relationship:
> $\mathring{\mathcal{F}}_F(L_x(\cdot), L_h(\cdot)) \subseteq \mathring{\mathcal{F}}_g(M_x, M_h)$.
>
> Assume that we have $F\in \mathring{\mathcal{F}}_F(L_x(\cdot), L_h(\cdot))$, where for some $M_x,M_h$:
> $$
>  L_x(t) = M_x \sqrt{\frac{1}{2 M_h}\left(e^{2M_h t} - 1\right)}; L_h(t) = e^{M_h t} - 1.
> $$
> We need to prove that the corresponding $\mathbf{g}$ is $M_x, M_h$-Lipschitz.
>
> **We start with the $x$ component.** Consider two input trajectories $\mathbf{x}_1 \ne \mathbf{x}_2$, and equal starting points $\mathbf{h}_1 = \mathbf{h}_2 = \mathbf{h}_0$. Since $F$ is Lipschitz, we can write the following inequality:
> $$
> \| \mathring{F}(\mathbf{x}_1, \mathbf{h}_0) - \mathring{F}(\mathbf{x}_2, \mathbf{h}_0) \| \triangleq \| \Delta \| \le \rho(\mathbf{x}_1, \mathbf{x}_2) M_x \sqrt{\frac{1}{2 M_h}\left(e^{2M_h t} - 1\right)}
> $$
>
> Here, we will also denote the difference between these trajectories by $\Delta$:
> $$
>     \mathring{F}(\mathbf{x}_1, \mathbf{h}_0) - \mathring{F}(\mathbf{x}_2, \mathbf{h}_0) = \Delta.
> $$
>
> We square the Lipschitz inequality, and take the derivative w.r.t. $t$:
> $$
>     2 (\Delta \mathbf{g}, \Delta) \le \frac{M_x^2}{2M_h} \left(\frac{d\rho^2}{dt} \left(e^{2M_h t} - 1\right) + \rho^2 \frac{d}{dt}\left(e^{2M_h t} - 1\right) \right).
> $$
> Here, we use the notation $\Delta \mathbf{g} = \mathbf{g}(\mathbf{x}_1(t), \mathbf{h}_1(t)) - \mathbf{g}(\mathbf{x}_2(t), \mathbf{h}_2(t)).$
> The derivatives of $\rho^2$ and $\left(e^{2M_h t} - 1\right)$ are easily calculated analytically:
> $$
> \frac{d\rho^2}{dt} = \| \mathbf{x}_1(t) - \mathbf{x}_2(t)\|^2.
> $$
> $$
> \frac{d}{dt} \left(e^{2M_h t} - 1\right) = 2M_h e^{2M_h t}.
> $$
> We substitute these expressions:
> $$
>     2 (\Delta \mathbf{g}, \Delta) \le \frac{M_x^2}{2M_h}
>     \left(
>         \| \mathbf{x}_1(t) - \mathbf{x}_2(t)\|^2. \left(e^{2M_h t} - 1\right)
>         +
>         \rho^2 2M_h e^{2M_h t}.
>     \right).
> $$
> Now we divide the above by $t$, and take the limit at $t \rightarrow 0$.
> $$
> \underset{t \rightarrow 0}{\lim} 2 (\Delta \mathbf{g}, \frac{1}{t} \Delta)
> \le
> \underset{t \rightarrow 0}{\lim} \frac{M_x^2}{2M_h}
>  \left(
>     \| \mathbf{x}_1(t) - \mathbf{x}_2(t)\|^2 \frac{1}{t} \left(e^{2M_h t} - 1\right)
>     +
>     \frac{1}{t} \rho^2 2M_h e^{2M_h t}
> \right)
> $$
> We calculate the limits for terms with $\frac{1}{t}$:
> $$
>     \underset{t \rightarrow 0}{\lim}\frac{1}{t} \Delta = \Delta \mathbf{g}(0);
> $$
> $$
>     \underset{t \rightarrow 0}{\lim} \frac{1}{t} \left(e^{2M_h t} - 1\right)
>     = 2M_h;
> $$
> $$
>     \underset{t \rightarrow 0}{\lim} \frac{1}{t} \rho^2 =
>     \| \mathbf{x}_1(0) - \mathbf{x}_2(0) \|^2.
> $$
> The limit for the other terms is achieved by simply setting $t=0$:
> $$
> 2 (\Delta \mathbf{g}(0), \Delta \mathbf{g}(0))
> \le \frac{M_x^2}{2M_h}
> \left(
>     \| \mathbf{x}_1(0) - \mathbf{x}_2(0)\|^2 2 M_h
>     +
>     \| \mathbf{x}_1(0) - \mathbf{x}_2(0) \|^2 2M_h
> \right).
> $$
> Simplifying, we get:
> $$
> \| \mathbf{g}(\mathbf{x}_1(0), \mathbf{h}_0) - \mathbf{g}(\mathbf{x}_2(0), \mathbf{h}_0) \|^2 \le M_x  \| \mathbf{x}_1(0) - \mathbf{x}_2(0)\|^2.
> $$
> Since the choice of $\mathbf{x}_1(0), \mathbf{x}_2(0), \mathbf{h}_0$ is arbitrary, the above implies that $\mathbf{g}$ is $M_x$-Lipschitz.
>
> **Now for the $h$ component**. We follow a very similar approach, but this time the input trajectories are the same $\mathbf{x}_1 = \mathbf{x}_2 = \mathbf{x}$, while the starting points are not $\mathbf{h}_1 \ne \mathbf{h}_2$.
> Therefore, since $F$ is Lipschitz, we can write the following inequality:
> $$
> \| \mathring{F}(\mathbf{x}, \mathbf{h}_1; t) - \mathring{F}(\mathbf{x}, \mathbf{h}_2; t) \| \triangleq \| \Delta(t) \| \le \|\Delta_0\| \left( e^{M_h t} - 1 \right).
> $$
> Here, $\Delta_0 \triangleq \mathbf{h}_1 - \mathbf{h}_2$. Again, we take the square and differentiate w.r.t. $t$:
> $$
> 2(\Delta, \Delta \mathbf{g}) \le \| \Delta_0 \|^2 2 \left( e^{M_h t} - 1 \right) M_h e^{M_h t}.
> $$
> Divide by $t$ and take the limit at $t \rightarrow 0$:
> $$
> 2 \| \Delta \mathbf{g} \|^2 \le  \| \Delta_0 \|^2 2 M_h.
> $$
> Since the choice of $\mathbf{x}, \mathbf{h}_1, \mathbf{h}_2$ is arbitrary, we have proven that $\mathbf{g}$ is $M_h$-Lipschitz.
>
> This concludes the proof of the new version of Theorem 3.1.

---

> > ### Comment · Reviewer_NLhn · 2025-11-26
> >
> > The reviewer appreciates the authors’ detailed response. The reviewer I understand that long-horizon integration accumulate the errors and the norm of hidden state $h_t$. However, my concern is not about universal approximation capability at fixed $T$ but rather about whether increasing the time horizon meaningfully enhances the model’s learnable expressivity once negative feedback is introduced.
> >
> > Theorem 3.1 is derived without negative feedback. This is critical, as the reviewer has provided an explicit example showing that introducing NF suppresses the internal evolution of the state as $T$ increases, which is an issue the authors have acknowledged. As a result, NF may completely outweigh the theoretical gain from increasing $T$.
> >
> > From a learning perspective, overly large $T$ may introduce stiffness, optimization difficulties, and collapsed dynamics. This suggests a practical trade-off: the time horizon is effectively a hyperparameter to tune rather than a dimension that can be increased arbitrarily to improve expressivity.
> >
> > Therefore, my evaluation remains the same.

---

> ### Author Response · Authors · 2025-11-27
>
> We hope that we have satisfied your first concern, regarding formal proof of time-scaling expressiveness, and your third concern, regarding the statistical significance of our results, since you do not bring them up in this comment.
>
> Now for the points you brought up in this comment:
>  - _"Theorem 3.1 is derived without negative feedback."_ Yes, but as we state in our rebuttal, _"Anti-NF provides a clean way to a) stabilize integration, but b) does not introduce any significant constraints on the resulting approximator."_ We underline, that the vector field of DeNOTS, Anti-NF, **does not constrain the trajectory**. For instance, the NF effect may be turned off altogether, by setting $a=1, b=0$, which still satisfies $a + b = 1$. As a result, we are left with $\frac{dh}{dt} = f_\theta$, which is precisely what Theorem 3.1 considered.
>  - _"the reviewer has provided an explicit example showing that introducing NF suppresses the internal evolution..."_ We did not quite grasp, why your example "shows that NF suppresses the internal evolution". On the contrary, as we state in our initial rebuttal, it shows "that Anti-NF may approximate virtually any function," as you demonstrate that it may model both exponential growth or exponential decay, even for such a basic setting, without an explicit dependence on $x$ and with an overly simplistic dependence on $h$. We would be grateful if you would explain your concern to us.
> - _"From a learning perspective, overly large $T$ may introduce stiffness, optimization difficulties, and collapsed dynamics. This suggests a practical trade-off: the time horizon is effectively a hyperparameter to tune rather than a dimension that can be increased arbitrarily to improve expressivity."_ We have **not observed such difficulties in our experiments**, which all show that increasing $T$ for Anti-NF increases the resulting quality, (the training routine is kept the same). Kindly, elaborate on that point: are there any specific theoretical difficulties you are referring to? That said, we are happy you agree that the time-scaling coefficient $D$ is a hyperparameter which needs tuning, which is one of our main claims. Specifically, we claim that a $T=1$ is not always optimal, and increasing it increases expressiveness. Scaling time may indeed bring diminishing returns, since increasing expressiveness is not always preferred for Machine Learning models.

---

### Official Review · Reviewer_7PJk · 2025-10-31

**Soundness:** 2
**Presentation:** 3
**Contribution:** 2
**Rating:** 6
**Confidence:** 3

**Summary:**

This paper proposes DENOTS (Deep Neural ODEs with Negative Feedback and Time Scaling), a novel neural ODE method for time series analysis. The authors introduce two main innovations: (1) time scaling to increase the number of function evaluations (NFE) without increasing weight norms, and (2) an anti-phase negative feedback (Anti-NF) mechanism to stabilize the dynamics. The paper claims that DENOTS achieves state-of-the-art performance on four public datasets. However, the experimental design is insufficient to fully validate the contributions, and the related work discussion is not comprehensive.

**Strengths:**

1.Novel Time Scaling Approach: The idea of scaling the integration time range to increase NFE without increasing weight norms is conceptually interesting and has potential to improve model expressivity.
2.Practical Performance: The experimental results demonstrate that DENOTS outperforms several baseline methods on multiple datasets, showing practical utility.
3.Stability Considerations: The inclusion of negative feedback mechanisms to address stability issues in neural ODEs is a thoughtful contribution.
4.Experimental Design: The paper includes experiments on multiple datasets and ablation studies to validate different components of the method.

**Weaknesses:**

1.Restrictive Assumptions: The theoretical analysis relies on restrictive assumptions (e.g., Assumptions 4.1 and 4.2) that may not hold in practice. The authors do not sufficiently discuss the rationality and limitations of these assumptions.
2.D/M Ratio Problem: The paper provides insufficient details on how to select D and M values, particularly regarding the determination of the D/M ratio. More specific parameter selection guidelines and analysis of how the D/M ratio affects model performance are needed.
3.Unclear Distinction from Existing Methods: The paper does not adequately explain the essential differences between the DENOTS method and existing GRU-ODE or other neural ODE variants. A clearer comparative analysis is needed.
4.Insufficient Ablation Study Analysis: It does not deeply analyze the interaction between time scaling and anti-phase negative feedback. The authors should further explore why the combination of these two techniques produces better performance.
5.Missing Computational Efficiency Analysis: The paper does not systematically compare computational efficiency with other methods, nor does it sufficiently discuss the computational cost incurred by increased NFE. A more comprehensive performance evaluation is needed.
6.Inadequate Related Work Discussion: The paper only briefly mentions that "prior work mostly sidesteps this topic" regarding time scaling, without citing specific related work. The discussion of negative feedback mechanism related work, particularly comparisons with methods like GRU-ODE-Bayes, is insufficient.

**Questions:**

1. How do you justify the restrictive assumptions (e.g., Assumptions 4.1 and 4.2) used in your theoretical analysis? Could you discuss their validity in practical scenarios?
2. Could you provide more details on how to select parameters D and M, particularly regarding the determination of the D/M ratio? How does this ratio affect model performance?
3. Could you provide a more detailed explanation of the anti-phase negative feedback mechanism? Specifically, why does passing -h instead of h to a standard PyTorch GRU solve the "forgetfulness" problem?
4. How does your method fundamentally differ from existing GRU-ODE or other neural ODE variants? Could you provide a detailed comparative analysis?
5. Could you provide a more in-depth analysis of the ablation study.Specifically, how do time scaling and anti-phase negative feedback interact to produce superior performance?
6. Could you provide a systematic comparison of computational efficiency with other methods? How do you balance the increased NFE with computational cost?
7. Could you provide a more comprehensive review of literature related to time scaling in neural ODEs? Are there any prior works that have explored similar ideas?
8. How does your negative feedback mechanism compare with existing methods like GRU-ODE-Bayes? Could you provide a detailed comparison?

---

> ### Author Response · Authors · 2025-11-21
>
> 1. Assumption 4.1 (Lipschitzness) is very common with Neural Network theory; for example, see [1] for an overview of this topic. Assumption 4.2 is checked in Appendix C.6.
> 2. The value of M is constant for each dataset, and is used to make inter-dataset values of D comparable. The value of D is responsible for increasing the NFE: the two are almost interchangeable, since a larger integration timeframe evidently requires more integration steps.
> To support this point, here is a table of correlations between NFE and D for various models on the Pendulum dataset:
> | Anti-NF | Sync-NF | ReLU | Tanh | No NF |
> |---------|---------|------|------|-------|
> | 0.93    | 0.96    | 0.87 | 0.78 | 0.94  |
>
> As for model performance, Figures 1, 8 and 9 illustrate and compare the performance of all our vector fields for varying values of NFE. It is our main claim, that by increasing D (thereby increasing NFEs), the performance of the Anti-NF model improves. Additionally, here is a figure, illustrating the (increasing) performance of the Anti-NF vector field on the Pendulum dataset: https://postimg.cc/NyT5tKrc.
>
> 3. As we state in the paper, negative feedback of the Anti-NF vector field is _less strict_ compared to that of Sync-NF. The difference is in the way these vector fields handle Assumption 4.2 (_NF Strength_), which is necessary for forgetfulness. We thoroughly analyze NF strength in Appendix C.6. For Sync-NF, the validity of said assumption depends only on the Lipschitz constant, fully determined by the learned weights. As we demonstrate in Figure 10, the learned weights always adhere to this assumption in practice. On the other hand, for Anti-NF Assumption 4.2 depends on a certain balance between $a,b$, both of which depend on the input. Consequently, the NF strength of Anti-NF may vary, i.e. the model may "choose" to reduce negative feedback for a specific input, reducing the forgetting effect.
> We directly test forgetfulness on the SineMix experiments (Table 4): we conclude that Anti-NF does not suffer from forgetfulness.
>
> We highlight, however, that in practice Anti-NF still enjoys all the benefits of Assumption 4.2. It is both stable and robust, as we demonstrate in the experiments of sections 4.1 and 4.2.
>
> 4. To the best of our knowledge, no prior methods intentionally upscale time beyond 0-1, aiming for better expressiveness. We are the first to introduce SNCDE (Scaled NCDE). As for the NF in GRU-ODE, we extensively compare with it both in theory and in practice throughout our paper (its SNCDE version is referred to as Sync-NF). We include an additional literature review paragraph below.
> 5. The ablation study is presented in sections 4.1, 4.2, 4.3. In short, without NF, the time-scaled (SNCDE) methods are unstable (4.1), and more sensitive to attacks (4.2), however, the NF from Sync-NF (scaled GRU-ODE) induces forgetfulness due to _strict_ negative feedback (4.3). Anti-NF does not possess any of these drawbacks, it may adapt the strength of its negative feedback (see point 3).
> 6. Computational cost scales linearly with increased NFE; NFE is often used as an estimate for computational costs. Such is often the case in Deep Learning: the more expressive models are more computationally-intensive. For model sizes, we refer you to Table 9: DeNOTS is the least parameter-intensive, since it just uses a single GRU cell.
> 7. To the best of our knowledge, we are the first to intentionally scale time, aiming for better expressiveness. We attach an additional Review paragraph below, considering the prior works related to time in greater detail.
> 8. See 3, 4, 5\.

---

> ### Author Response · Authors · 2025-11-21
>
> **Review of Time in Neural ODE (4, 7)**. To escape the instability, introduced by possibly large time frames, most prior works enforce time to be roughly 0-1.
> The earliest works normalize it directly [2, 3, 4], dividing each distinct timeframe from the data by its duration.
> Future work proposes a reparametrization, so that the time scale does not influence the resulting model [5].
> This is similar to the way Neural CDEs handle time [6], they are also provably insensitive to the speed at which the data is traversed.
> Finally, [7] is probably the closest anyone has gotten to intentionally choosing the integration timeframe, albeit on non-sequential data.
> The authors propose to predict the time of integration.
> However, the motivation is significantly different: the predicted time is bounded, meaning that they only allow to cut the integration short.
> Early stopping of integration allows the authors to add another degree of freedom when considering low-dimensional hidden states, therefore in a way increasing the expressivity of the model.
> On the other hand, we work with relatively high-dimensional hidden trajectories to escape these limitations, we upscale the integration interval (instead of stopping the integration short), following a different, Lipschitz-inspired idea, and, most importantly, we work with time-series data.
> A large portion of our ideas are meaningless, if we consider non-temporal data.
> All in all, to the best of our knowledge, there are no existing works that consider time to be an important hyperparameter.
>
> ### References:
>  - [1]: Virmaux, Aladin, and Kevin Scaman. "Lipschitz regularity of deep neural networks: analysis and efficient estimation." Advances in Neural Information Processing Systems 31 (2018).
>  - [2]: Ricky TQ Chen, Yulia Rubanova, Jesse Bettencourt, and David K Duvenaud. Neural ordinary
> differential equations. Advances in NeurIPS, 31, 2018
>  - [3]: Yulia Rubanova, Ricky TQ Chen, and David K Duvenaud. Latent ordinary differential equations for irregularly-sampled time series. Advances in NeurIPS, 32, 2019.
>  - [4]: Emilien Dupont, Arnaud Doucet, and Yee Whye Teh. Augmented neural odes. Advances in neural
> information processing systems, 32, 2019.
>  - [5]: Ricky TQ Chen, Brandon Amos, and Maximilian Nickel. Neural spatio-temporal point processes.
> arXiv preprint arXiv:2011.04583, 2020
>  - [6]: Patrick Kidger, James Morrill, James Foster, and Terry Lyons. Neural controlled differential equations
> for irregular time series. Advances in NeurIPS, 33:6696–6707, 2020.
>  - [7]: Stefano Massaroli, Michael Poli, Jinkyoo Park, Atsushi Yamashita, and Hajime Asama. Dissecting
> neural odes. Advances in Neural Information Processing Systems, 33:3952–3963, 2020b.

---

### Meta-Review · Area_Chair_Z7ZL · 2025-12-31

**Summary:**

This work introduced DENOTS, a novel neural ODE method for time series analysis. It introduces two main components: 1) increase the time horizon of the differential equation to increase expressivity; and 2) add a negative feedback term proportional to the current state, to avoid instability. The extensive experimental results demonstrated the good performance of the proposed DENOTS.


Strength:

1. The idea is conceptually simple, yet seems novel and sound.
2. The proposed two main components can enhance both expressivity and stability.
3. The evaluation results demonstrated that DENOTS outperforms several baseline methods on multiple datasets.


Limitations:

1. One concern is that scaling T would also 'exponentially' impair the model's performance. While authors have shown the proof during rebuttal, it still needs to consider the negative feedback.

2. Part of the experimental results are not significantly compared to baselines


In summary, the authors have addressed most concerns raised by the reviewers. The proposed method is interesting and technically sound. I tend to recommend accepting this work.

**Reviewer Concerns:**

For reviewer 7PJk, the authors addressed the concerns about the connection between NFE and D, and the effect of D.

For reviewer NLhn's concern about Theorem 3,  the authors proved this more general version of the theorem.

The authors also dealt with concerns about numerical error from Reviewer PDGR

**Reviewer Scores:**

I think both reviewers 7PJk and NLhn may increase their scores since the authors addressed their main concerns. Please see the summary in the above Reviewer concerns.

---

### Decision · Program_Chairs · 2026-01-26

Accept (Poster)